# Multi-heme cytochrome-mediated extracellular electron transfer by the anaerobic methanotroph 'Candidatus Methanoperedens nitroreducens'

Xueqin Zhang[1], Georgina H. Joyce[2], Andy O. Leu[2], Jing Zhao[1,3], Hesamoddin Rabiee[1,4,5], Bernardino Virdis[1], Gene W. Tyson[2], Zhiguo Yuan[1,6], Simon J. McIlroy[2] & Shihu Hu[1] ✉

Anaerobic methanotrophic archaea (ANME) carry out anaerobic oxidation of methane, thus playing a crucial role in the methane cycle. Previous genomic evidence indicates that multi-heme c-type cytochromes (MHCs) may facilitate the extracellular electron transfer (EET) from ANME to different electron sinks. Here, we provide experimental evidence supporting cytochrome-mediated EET for the reduction of metals and electrodes by 'Candidatus Methanoperedens nitroreducens', an ANME acclimated to nitrate reduction. Ferrous iron-targeted fluorescent assays, metatranscriptomics, and single-cell imaging suggest that 'Ca. M. nitroreducens' uses surface-localized redox-active cytochromes for metal reduction. Electrochemical and Raman spectroscopic analyses also support the involvement of c-type cytochrome-mediated EET for electrode reduction. Furthermore, several genes encoding menaquinone cytochrome type-c oxidoreductases and extracellular MHCs are differentially expressed when different electron acceptors are used.

Anaerobic oxidation of methane (AOM) is an essential microbiological process in the global methane cycle that controls net methane emissions from anoxic subsurface environments both in marine and freshwater habitats[1]. It is predominantly mediated by several uncultivated lineages of anaerobic methanotrophic archaea (ANME) that are phylogenetically related to methanogens[2]. Reverse methanogenesis remains the prevailing metabolic hypothesis for AOM, which is thermodynamically favourable when coupled to the reduction of suitable electron acceptors. To date, described terminal electron acceptors for AOM include sulfate[2,3], nitrate[4], metal oxides[5–7], and humic substances[8,9].

Nitrate-dependent AOM is well-characterised in several species of the freshwater ANME-lineage *Methanoperedenaceae*, and is mediated by an encoded nitrate reductase[4,10]. Meanwhile, the biochemical mechanisms that ANME employ for the reduction of other terminal electron acceptors are less understood. Sulfate-dependent AOM is carried out by marine ANME-lineages in syntrophic partnership with sulfate-reducing bacteria (SRB). Studies using microcosm experiments, metagenomics and theoretical modelling refute the syntrophic association by diffusive exchange of intermediates (such as hydrogen, formate, acetate, methanol) among ANME-SRB consortia but reconcile the diffusion-independent direct electron transfer mechanism[8,11–13].

[1]Australian Centre for Water and Environmental Biotechnology (ACWEB), Faculty of Engineering, Architecture and Information Technology, University of Queensland, Brisbane, Australia. [2]Centre for Microbiome Research, School of Biomedical Sciences, Queensland University of Technology (QUT), Translational Research Institute, Woolloongabba, Australia. [3]Ecological Engineering of Mine Wastes, Sustainable Minerals Institute, The University of Queensland, Brisbane, QLD, Australia. [4]School of Chemical Engineering, The University of Queensland, Brisbane, QLD, Australia. [5]Centre for Future Materials, University of Southern Queensland, Springfield, QLD, Australia. [6]School of Energy and Environment, City University of Hong Kong, Hong Kong SAR, China. ✉e-mail: s.hu@uq.edu.au

Genetic analysis and redox-dependent staining of marine ANME-SRB assemblages suggest that cytochrome proteins provide the biochemical foundations for direct electron transfer from ANME to SRB within the consortia[13]. Metagenomic analysis has indicated that *Methanoperedenaceae* may employ cytochromes for electron transfer to metal oxides during AOM. This hypothesis was supported by the high expression of multi-heme *c*-type cytochromes (MHCs) encoded by these species when enriched with iron or manganese oxides as the terminal electron acceptors[6,7]. It has also been postulated that ANME may use an equivalent mechanism to reduce humics and their analogues[8,9]. Thus, cytochromes are hypothesized to mediate the reduction of electron acceptors for many AOM processes by ANME in the *Methanoperedenaceae* family, however, empirical evidence is still missing to support the hypothesis.

The cytochrome-based electron transfer machinery has been rigorously characterised for a range of phylogenetically diverse microorganisms, with *Geobacter* and *Shewanella* as the best-studied examples[14]. Both *Geobacter* and *Shewanella* can respire a variety of electron acceptors, which is largely attributed to *c*-type cytochromes they encode[15,16]. Holmes and colleagues recently used *Methanosarcina acetivorans* as a genetically tractable model microbe to verify that the methanogenic relative of ANME can use membrane-bound cytochrome as a conduit to transfer electrons to an extracellular electron acceptor[17], which has a mechanistic implication for the extracellular electron transfer (EET) of ANME. MHC genes are more common and abundant within ANME lineages relative to related methanogens[18,19]. However, investigation of the EET mechanisms of ANME has been stymied by the lack of an ANME pure culture. The current knowledge of cytochrome-based EET pathways of ANME is relatively poor and predominantly inferred from metagenomic and metatranscriptomic prediction[6,7,20]. To date, in vivo identification, and biochemical and biophysical characterization of MHCs in ANME beyond genomic and transcriptomic analyses are still lacking to verify that cytochromes function as conduits to transfer electrons across the outermost layer to the cell exterior[18].

In this study, we examine the role of cytochromes in EET behaviours of the ANME species '*Candidatus* Methanoperedens nitroreducens'. This species has been shown to perform nitrate-dependent AOM, but is also capable of catalysing the reduction of other extracellular electron acceptors such as iron oxide and biochar, using mechanisms that remain unexplored[5,21,22]. Here, we report that '*Ca*. M. nitroreducens' can transfer electrons to metals (including soluble iron and silver) and poised electrodes. Using a combination of fluorescence and electron microscopic, electrochemical, spectroscopic and bioinformatic methods, our findings show that MHCs are central to EET mechanisms of '*Ca*. M. nitroreducens'. Differential expression of specific MHCs upon substitution of the available electron acceptor indicates that '*Ca*. M. nitroreducens' shift between multiple MHC-mediated electron pathways, allowing for the observed metabolic flexibility of this ANME species.

## Results and discussion

### Fluorescence visualization and metatranscriptomic analysis reveal the role of MHCs in direct EET for '*Ca*. M. nitroreducens'

Previous studies have demonstrated that '*Ca*. M. nitroreducens' enrichment culture, which had been acclimated to nitrate reduction, was able to catalyse iron reduction[5,22]. Whether '*Ca*. M. nitroreducens' can reduce iron independently and what EET mechanism it employs to do so remains unknown. High expression of MHCs was previously reported by the related species '*Ca*. Methanoperedens ferrireducens' during iron reduction[6]. As '*Ca*. M. nitroreducens' also encodes a number of MHC genes, it was speculated that it may also express these for direct EET pathways for iron reduction[22]. To test this hypothesis, we compared gene expression for a '*Ca*. M. nitroreducens'-dominated culture with either nitrate or iron as the sole terminal electron

acceptor, and a series of characterisations on MHCs were conducted. The AOM activities observed (based on $^{13}CO_2$ production rates from fed $^{13}CH_4$) when incubated with soluble iron (ferric citrate) or nitrate were similar (Fig. 1a, Supplementary Fig. 1). $Fe^{3+}$ reduction was simultaneously observed with $^{13}CO_2$ production at a stoichiometry close to the expected ratio of 1:8 (Fig. 1a), which confirmed iron-dependent AOM.

Given that the observed AOM could be performed by the '*Ca*. M. nitroreducens' via EET directly to iron, or to iron-reducing syntrophic partners[21,23], FeRhoNox fluorescence staining was employed to identify the microorganisms directly responsible for the observed iron reduction. The method is based on $Fe^{2+}$-selective chemosensory fluorescence on the cell surface of active iron-reducing microorganisms[24]. Under iron-reducing incubations, '*Ca*. M. nitroreducens' cells exhibited bright FeRhoNox fluorescence (Fig. 1b, c), indicating that $Fe^{2+}$ was absorbed to their outer surface[24]. FeRhoNox positive cells almost entirely overlapped with '*Ca*. M. nitroreducens' (Fig. 1d; congruency = $94.1 \pm 2.5$ %, $n = 20$), while cells of other populations in the consortium showed fluorescence similar to the negative control (Supplementary Fig. 2g, j), suggesting the former population was responsible for the bulk of the observed iron reduction in the system (Supplementary Fig. 2e–j). Giving confidence in the specificity of the method, FeRhoNox positive signal was not observed for (1) the $Fe^{3+}$-amended '*Ca*. M. nitroreducens' enrichment culture that was immediately stained with FeRhoNox for visualization (i.e., without incubation process for $Fe^{3+}$ reduction) (Supplementary Fig. 2a, b), or for (2) '*Ca*. M. nitroreducens' enrichment culture that was incubated for $Fe^{3+}$ reduction but in the absence of staining (Supplementary Fig. 2c, d). Collectively, these results indicate that iron reduction was performed by '*Ca*. M. nitroreducens' cells via direct EET.

To better understand the metabolic pathway of iron-dependent AOM by '*Ca*. M. nitroreducens', comparative metatranscriptomic analyses was performed across the nitrate and iron-fed batch conditions. Metagenomic results showed a similar microbial community structure under both conditions, with '*Ca*. M. nitroreducens' being the most dominant population both in the nitrate-reducing incubation (38.7%) and in the iron-reducing incubation (32.2%) (Fig. 1e; Supplementary Data 1). Metatranscriptomic analysis revealed that the '*Ca*. M. nitroreducens' population also dominated the gene expression profiles under both conditions (91.9% and 89.0% of the total mRNA for the nitrate-reducing and iron-reducing incubation, respectively (Fig. 1e; Supplementary Data 1), suggesting they were responsible for the bulk of nutrient transformations in each batch incubation.

Consistent with evident methane oxidation activities (Fig. 1a and Supplementary Fig. 1), '*Ca*. M. nitroreducens' showed high expression of the AOM pathway under both electron acceptor conditions (Fig. 1f, Supplementary Data 2, 3). The *narGHI* operon, which encodes a membrane-bound nitrate reductase complex (Supplementary Data 3), was highly expressed under the nitrate-fed condition and dramatically downregulated when nitrate was replaced with iron (Fig. 1f, Supplementary Data 2). In contrast, MHC expression by '*Ca*. M. nitroreducens' was significantly higher when iron replaced nitrate supplementation (Fig. 1f, Supplementary Data 2, 6), supporting the importance of these proteins for observed iron reduction. The annotated MHC genes expressed were predominantly encoded by the '*Ca*. M. nitroreducens' (Fig. 1f, Supplementary Data 5), indicating that this species reduces iron independent of a syntrophic partner organism.

### Single-cell visualization supports metal reduction mediated by surface-localized cytochromes of '*Ca*. M. nitroreducens'

To further test the hypothesis that '*Ca*. M. nitroreducens' uses cytochromes as electron transfer conduits for EET, the potential localisation of extracellular cytochromes was firstly visualized by TEM through cytochrome reactive staining with 3,3'-diaminobenzidine (DAB)[13,20,25]. Treatment of the '*Ca*. M. nitroreducens' enrichment

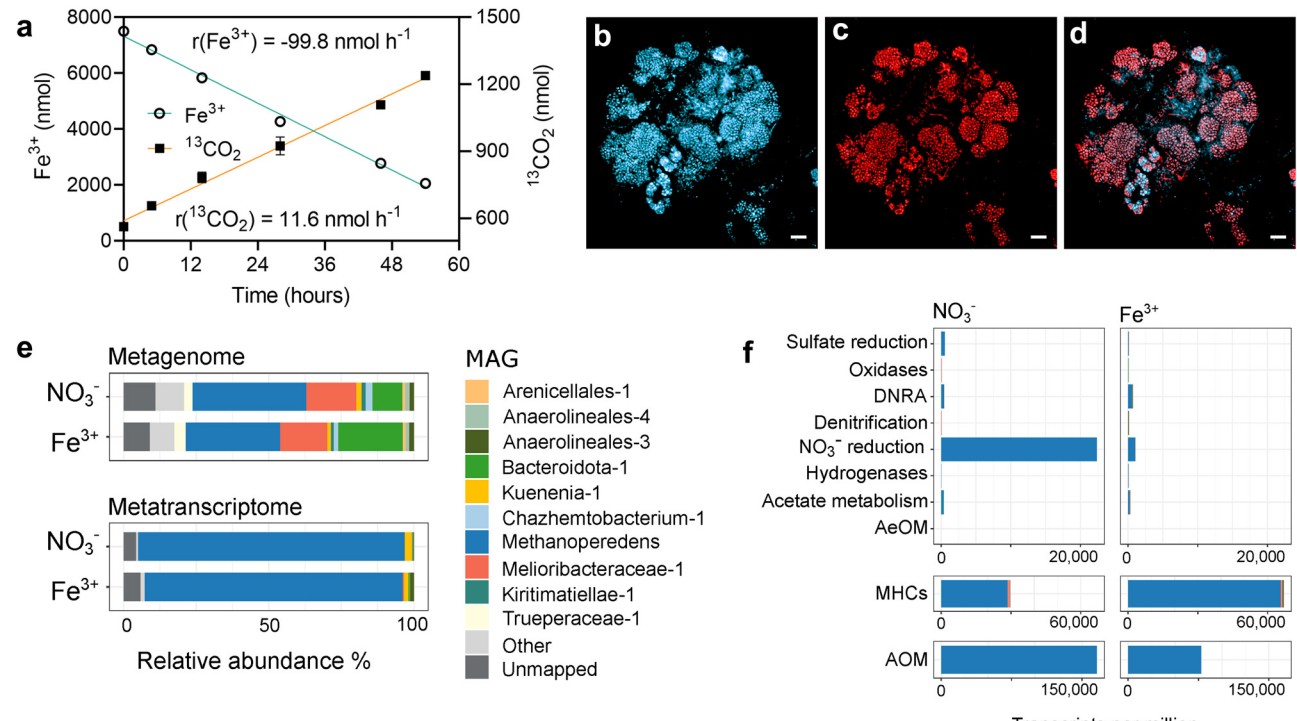

**Fig. 1 | Iron-dependent AOM catalysed by 'Ca. M. nitroreducens' enrichment culture. a** $Fe^{3+}$ reduction and $^{13}CO_2$ production (from $^{13}CH_4$ oxidation) during the iron-reducing incubation. Results from three independent biological replicates are represented as mean ± SD. Source data are provided as a Source Data file. **b–d** Fluorescent characterization of iron reduction. The experiment was repeated independently three times with similar results. **b** Autofluorescence image of methanogenic cofactor $F_{420}$ representing 'Ca. M. nitroreducens' cells. **c** Fluorescence microscopy image of cells stained with $Fe^{2+}$-specific FeRhoNox representing iron-reducing cells in the consortium. **d** Overlay of fluorescent image (**c**) and (**d**). Scale bars, 10 μm. **e** Metagenomic and metatranscriptomic community profiles for the iron-reducing incubation and nitrate-reducing incubations. Charts display the average of three biological replicates, except the nitrate-fed

metagenome chart, which represents the nitrate-fed seed bioreactor from which all batches were inoculated. **f** Community gene expression of selected metabolic pathways. Charts display the average of three biological replicates (Supplementary Data 3). 'Acetate metabolism' includes genes, *pta, acs, ack, acdA*. 'Sulfate reduction' includes dissimilatory sulfate reduction genes, *sat* and *aprA*. 'Oxidases' includes cytochrome c oxidase genes, *ccoN, ccoO, ccoP, coxA, coxB*. 'DNRA' is the abbreviation for 'Dissimilatory nitrate reduction to ammonium'. 'AeOM' is for 'Aerobic methane oxidation' and includes *mmo* monooxygenase genes. 'AOM' is for 'Anaerobic oxidation of methane' and includes *mcr* reverse methanogenesis genes. MHCs were predicted by identifying proteins with ≥ 3 CXXCH amino acid motifs. 'Other' includes MAGs at less than 1% relative abundance in both the metagenome and metatranscriptome. **e** and **f** share the same legend.

culture following iron-reducing incubation with DAB and $H_2O_2$ resulted in intense staining of the outermost layer of 'Ca. M. nitroreducens' (Fig. 2a, b), while no staining was visible for cells treated with DAB alone (Fig. 2c, d). Similar staining observations have also been reported for EET-capable bacteria of *S. oneidensis* and *G. sulfurreducens*[25], as well as for ANME that were implied to use MHCs as redox-active proteins to interact with SRB[13,20]. Moreover, 'Ca. M. nitroreducens' cells from the iron-reducing incubation exhibited more intense staining on the cell surface than that from the nitrate-reducing control (Fig. 2a–d; Supplementary Fig. 3a–d), in line with the higher expression of MHCs when reducing iron (Fig. 1f). Overall, the DAB positive staining supported that cell-surface associated cytochromes are important for respiratory iron reduction by 'Ca. M. nitroreducens'.

The nitrate-fed 'Ca. M. nitroreducens' enrichment culture was also incubated with Ag (I) to examine whether they can also mediate extracellular Ag (I) reduction. It has been demonstrated that EET-capable microorganisms can enzymatically reduce Ag (I) via the mechanism involving *c*-type cytochromes, resulting in precipitation of nanoscale Ag (0) on the cell surface[26,27]. As 'Ca. M. nitroreducens' cells can be distinguished from other populations based on their size and spatial arrangement, examining the potential deposition of the AgNPs on their cell surface would further support their ability for direct EET for metal reduction. After a 12 h incubation, the colour of the solution turned brown

(Supplementary Fig. 4a, c, d), indicative of silver nanoparticle (AgNPs) formation[27]. Thin sectioning TEM showed AgNPs were deposited either on outer envelope surface or associated-extracellular polymeric substances (EPS) of 'Ca. M. nitroreducens' cells (Fig. 2e, f; Supplementary Fig. 5a, b, e). The observation of AgNPs on the outer surface of cells are similar to the precipitation of Ag (0) on the outer membrane of *G. sulfurreducens* and *Thermincola potens* from their respiratory Ag (I) reduction[26,27]. Meanwhile, as the EPS of EET-capable microorganisms is an electroactive transient media to transfer electrons outward across the EPS layer[28], it is rational to see the precipitation of AgNPs also on the surface of EPS. In comparison, the addition of acetyl methionine (AcMet) to the incubation inhibited AgNPs formation (Supplementary Fig. 4b) and the deposition of Ag (0) on the outmost surface of 'Ca. M. nitroreducens' cells (Fig. 2g, h; Supplementary Fig. 5c, d). The axial ligand coordination reaction of the *c*-hemes with AcMet has been demonstrated to increase the reduction potential ($E^{0'}$) of *c*-type cytochromes to cause the formation of a large energy barrier for the electron transfer process[29,30]. This AcMet coordination could thus block the electron flow through electron transport chains of 'Ca. M. nitroreducens' for the observed inhibition of silver reduction. These results collectively suggest that 'Ca. M. nitroreducens' uses direct EET pathways for Ag (I) reduction to Ag (0) with *c*-type cytochromes localized on the cell surface or anchored to EPS surface as electrical conduits.

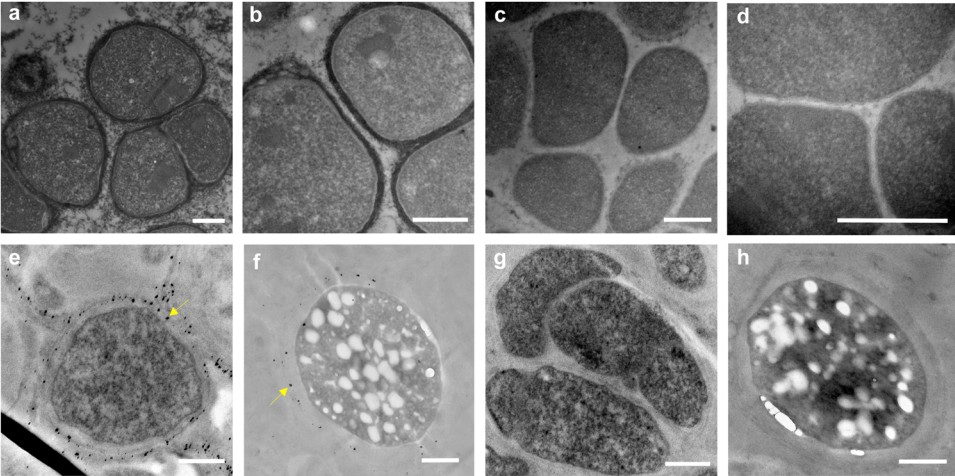

**Fig. 2 | TEM visualization of the 'Ca. M. nitroreducens' MHCs in the metal-reducing incubations. a–d** TEM images showing heme reactivity of 'Ca. M. nitroreducens' cells (identified by their distinctive morphology and size) in the iron-reducing incubation. Three independent experiments were conducted with similar results. **a, b** Positive DAB staining in the presence of $H_2O_2$. **c, d** Negative DAB staining in the absence of $H_2O_2$. **e–h** TEM images showing silver nanoparticle (AgNPs) deposition in the Ag (I)-reducing incubation. Six independent experiments were conducted with similar results. **e, f** AgNPs deposition (indicated by arrows) on the outer surface of 'Ca. M. nitroreducens' cells. **g, h** Absence of AgNPs deposition due to inhibition of cytochromes by Acetyl methionine (AcMet). Scale bars, 500 nm.

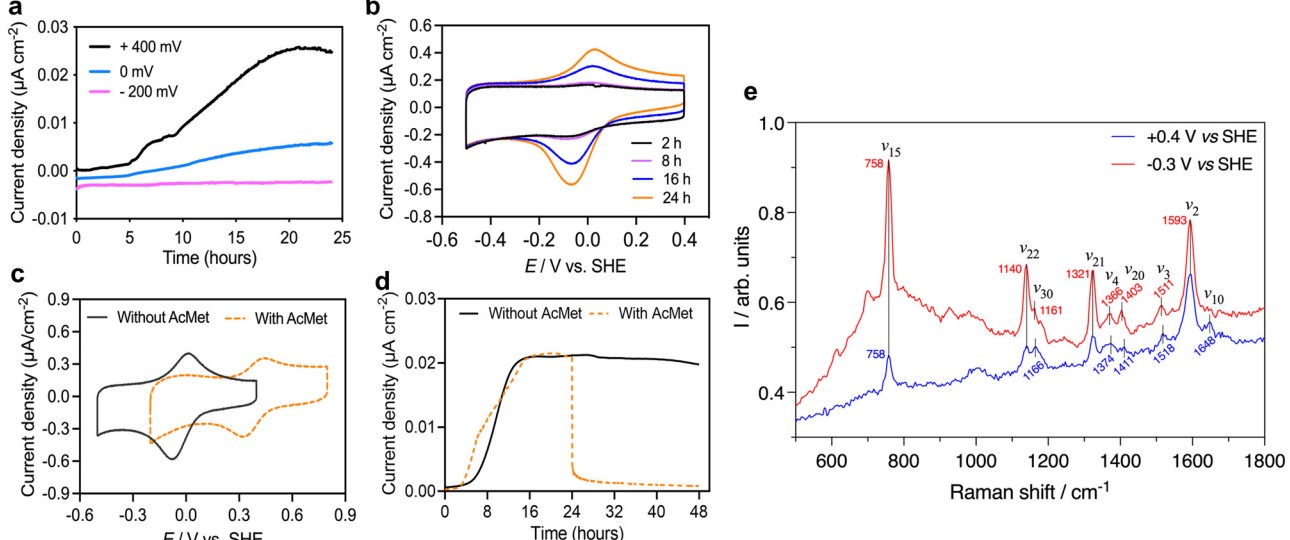

**Fig. 3 | Electrochemical and spectroelectrochemical characterization of 'Ca. M. nitroreducens'. a** Representative anodic current responses of the 'Ca. M. nitroreducens' enrichment culture to different poised potentials when respiring on fluorine-doped tin oxide (FTO) electrodes. **b** Whole-cell CVs of the electrochemically incubated (at +0.4 V vs SHE) 'Ca. M. nitroreducens' enrichment culture at different incubation times. **c** The effect of acetyl methionine (AcMet) treatment on whole-cell CVs of the 'Ca. M. nitroreducens' enrichment culture. **d** Representative anodic current of the 'Ca. M. nitroreducens' enrichment culture with and without AcMet treatment. **e** Representative resonance Raman spectra of the 'Ca. M. nitroreducens'-dominant biofilm acquired at electrochemically manipulated oxidative (blue) and reductive (red) status. Source data are provided as a Source Data file.

## Electrochemical and in situ spectroelectrochemical analysis supports cytochrome-mediated electrode reduction

Several studies have provided evidence that enriched 'Ca. M. nitroreducens' can respire on solid electrodes[31–33]. To provide evidence that cytochromes mediate the EET of 'Ca. M. nitroreducens' for electrode reduction, we carried out three separate experiments, including (1) the electrochemical characterization of 'Ca. M. nitroreducens' culture on fluorine-doped tin oxide (FTO) electrodes; (2) long-term incubation of 'Ca. M. nitroreducens' enrichment culture in bioelectrochemical systems (BESs), which allowed the metatranscriptomic analysis on ANME's EET pathway when respiring on solid electrodes; (3) application of Raman microspectroscopy for the real-time characterization of the redox states of c-type cytochromes of electrode-associated biofilms enriched in 'Ca. M. nitroreducens'.

When FTO electrodes poised at 0.4 V (vs. SHE) served as the sole electron acceptor (Supplementary Fig. 6), anodic current density was observed to gradually increase to just under 0.03 μA cm$^{-2}$ in 24 h (Fig. 3a). Cyclic voltammetry (CV) of the anodes with microbial inoculum exhibited a reversible redox couple (Fig. 3b), while no redox peaks were observed in the abiotic control (Supplementary Fig. 7a), further demonstrating that the 'Ca. M. nitroreducens' enrichment culture was capable of electron exchange with the electrode. This was supported by the peak currents in the recorded CVs at different incubation times, which increased over time (Fig. 3b), aligning with the

increase of chronoamperometric current (Fig. 3a). No redox peaks were observed on the voltammograms obtained with filtered (0.22 μm) cell-free media (Supplementary Fig. 7a), suggesting the electrochemical response of the culture was not due to soluble mediators. Plots of the peak current densities as a function of the scan rate displayed a linear relationship (Supplementary Fig. 7b, c), indicating adsorption of the redox-active enzymes on the electrode surface, and supporting the hypothesis that the 'Ca. M. nitroreducens' enrichment culture is capable of direct electron exchange with the electrodes[34,35]. This was in line with the physical attachment of 'Ca. M. nitroreducens' cells on the FTO electrode surface (Supplementary Fig. 7d).

Analysis of the cyclic voltammograms indicates a reversible redox-active site centred at a formal potential $E^f$ of ca. −20 mV vs SHE, as determined by the arithmetic average of the oxidative and reductive peaks. These observations are similar to previous observations of *Shewanella* spp. electroactive biofilms where outer membrane *c*-type cytochromes (OMCs) were identified as the redox active components[34,36]. Experiments performed under continuous electrode polarisation at a potential higher than −20 mV yielded current responses and obvious voltammetric redox peaks, whereas no current response or distinct CV redox peaks were observed when the potential was poised at levels lower than −20 mV (Fig. 3a, Supplementary Fig. 8), suggesting that no metabolic energy was available for microbial growth and biofilm development under this seeming thermodynamic barrier. This is in line with the fact that electroactive microbes require the electrode to be poised at a sufficiently positive potential to engage with the EET pathway[37]. To further examine whether the redox-active species governing the direct EET pathway of 'Ca. M. nitroreducens' enrichment culture were putative *c*-type cytochromes, we performed an axial-coordination reaction on the heme groups of *c*-type cytochromes under living conditions by using the specific binding affinity of AcMet. The $E^f$ shifted to around +390 mV (Fig. 3c), in contrast to −20 mV observed prior to the AcMet addition. The increase of redox potential as a response to AcMet coordination was also previously observed for the purified *c*-type cytochrome proteins[29], and OMCs in living electroactive *Shewanella* cells[30]. It caused the formation of a large energy barrier for the electron-exchange process of electroactive bacteria (*Shewanella and Geobacter*)[30,38]. Congruously, the addition of 100 mM AcMet also resulted in complete suppression of the catalytic current density of the 'Ca. M. nitroreducens' enrichment culture incubated at 0.4 V vs SHE (Fig. 3d). These results combined suggest that *c*-type cytochromes governed the EET of 'Ca. M. nitroreducens' enrichment culture to directly interact with electrodes.

To further understand the metabolic pathway of electrode-dependent AOM, 'Ca. M. nitroreducens' enrichment was cultivated for respiratory electrode reduction for long-term (72 days), and metatranscriptomic analysis was performed with biofilm formed on the electrodes of BESs (Supplementary Fig. 9). The current generation (Supplementary Fig. 10a) was found to be correlated with methane oxidation (Supplementary Fig. 10b) (according to $^{13}CO_2$ accumulation from $^{13}CH_4$ oxidation), confirming the occurrence of electrode-dependent AOM. Metatranscriptomic analysis revealed that 'Ca. M. nitroreducens' and *Geobacter anodireducens* were the most transcriptionally active populations in the community (9.6% and 23.6% of total mRNA reads, respectively) at the time of sampling (Supplementary Fig. 11a, b and Supplementary Data 1). The community showed negligible transcriptomic activities for dissimilatory sulfate and nitrate reductase genes, whilst it importantly showed high expression of MHCs for EET (Supplementary Fig. 11b), which was consistent with the current generation by the consortium (Supplementary Fig. 10a).

In the biofilm community, we observed the high expression of MHCs for 'Ca. M. nitroreducens', along with the known electroactive taxa *Geobacter anodireducens* and *Ignavibacterium* sp.[32,39]. As neither of the latter populations encoded methane-metabolising pathways, they may source carbon from diffusible methane-derived intermediates from 'Ca. M. nitroreducens' and/or cellular detritus. Acetate was previously identified as a likely key intermediate linking electron transfer between methane-oxidizing archaea and electrode-reducing bacteria[40,41]. In these previous studies, the methanotrophic cells were largely present in their planktonic form, owing to their suggested diffusion-dependent EET pathway by using acetate as a diffusive electron intermediate, rather than a contact-based EET pathway[40,41]. Conversely, in the current study all biomass was present as a biofilm attached to the electrodes (Supplementary Fig. 12a–d), with planktonic cells rarely observed by microscopy and detectable DNA/RNA being not retrievable from the supernatant. Furthermore, the $^{13}C$ isotopic tracer for methane metabolic flux analysis in our previous study of these ANME-enrichment BESs revealed that detected acetate was not the direct product of AOM but of fermentative degradation of intracellular or extracellular polymeric substances[33]. Fermentation biproducts likely provide a carbon and electron source for the non-methanotrophic electroactive bacteria in the biofilm community, which was supported by observed gene expression of acetate metabolism pathway in these species (mainly *Geobacter*) (Supplementary Fig. 11b, Supplementary Data 4). This was consistent with the current efficiency results showing that faradic contribution to the observed current generation was only partially attributable to AOM (Supplementary Fig. 13) as observed for other studies[31,32,39]. In addition, FISH analysis did not show an obligate proximal association between the 'Ca. M. nitroreducens' and the *Geobacter* populations (Supplementary Fig. 12a), which would have supported direct interspecies electron transfer (DIET) between the two. Thus, 'Ca. M. nitroreducens' likely interacted with electrodes independent of *Geobacter*. As *Geobacter* was not detected in the inoculum biomass used for the short-term electrochemical characterizations (Fig. 1e), and it appeared only after long-term bioelectrochemical incubations (Supplementary Fig. 11a), the *c*-type cytochromes identified from the short-term in vivo electrochemistry analyses (Fig. 3b–d) most likely originated from 'Ca. M. nitroreducens'. The in vivo electrochemical identification and characterization of *c*-type cytochromes combined with transcriptomic analyses all support the hypothesis that 'Ca. M. nitroreducens' is able to use MHC-based direct electron transfer pathways to drive electrode-dependent AOM.

To investigate the effective coupling of the electrode with *c*-type cytochromes associated to 'Ca. M. nitroreducens', we examined the characteristic resonance Raman spectra collected from biofilms under continuous polarisation conditions[42,43]. To allow for in vivo measurements, we used an electrochemical cell specifically designed to enable simultaneous collection of scattering information whilst applying a fixed electrode potential (Supplementary Fig. 14c). Biofilm formation of 'Ca. M. nitroreducens' associated with the surface of the electrode was again confirmed by fluorescence microscopic visualization (Supplementary Fig. 15a). As confirmed by CV analysis discussed above (Fig. 3b), during collection of the Raman spectra, the electrode was poised at either +0.4 V or −0.3 V *vs*. SHE to allow full oxidation and full reduction, respectively, of the adsorbed redox species responsible for EET in 'Ca. M. nitroreducens'. Prior to collection of the Raman spectra, each potential was maintained for at least 15 min to allow the establishment of electrochemical equilibrium between the electrode and the redox active moiety, as indicated by a stable current *versus* time profile (Supplementary Fig. 15b). Resonance Raman spectra collected under steady-state polarisation conditions are reported in Fig. 3e together with band assignment. Under both reducing and oxidising electrochemical conditions, the collected spectra presented vibrational bands typical of *c*-type cytochromes: $\nu_{15}$ at 758 cm⁻¹, $\nu_{22}$ at 1140 cm⁻¹, $\nu_{21}$ at 1321 cm⁻¹, $\nu_2$ at 1593 cm⁻¹, with additional minor bands in the mid-frequency region (1300–1700 cm⁻¹), which are also associated to vibrational modes of cytochrome heme groups: $\nu_4$, $\nu_{20}$, $\nu_3$, and $\nu_{10}$[44]. Under oxidising conditions (i.e. at +0.4 V), these bands were centred at 1374, 1411, 1518, and 1648 cm⁻¹, respectively. The application

of reducing conditions (−0.3 V) resulted in an increase in overall spectra intensity, with the bands $v_4$, $v_{20}$ and $v_3$ downshifting to 1366, 1403, and 1511 cm$^{-1}$, and the $v_{10}$ band no longer resolvable. In the low-frequency region, the band $v_{15}$- attributed to the pyrrole breathing mode−increased significantly in intensity. The intensity of this band is proportional to the number of hemes in the reduced states[45–48]. All these changes in spectral features are consistent with a $c$-type heme group having a six-coordinated low-spin state central iron atom shifting redox-status from an oxidised state (at 0.4 V) to a reduced state (at −0.3 V)[45,49]. As no other known electroactive bacteria was present in the '$Ca$. M. nitroreducens'-dominated biofilm consortium (Supplementary Fig. 15c), resonance Raman results collectively support the hypothesis that the direct EET pathway of '$Ca$. M. nitroreducens' is mediated by $c$-type cytochromes.

## Comparative expression analyses of putative MHCs for '*Ca*. M. nitroreducens' when respiring with different terminal electron acceptors

To gain further insight into the MHC-based EET pathways of '$Ca$. M. nitroreducens', transcriptional responses of its putative MHC genes were compared for incubations with different electron acceptors, including nitrate, soluble iron and long-term on a solid electrode. '$Ca$. M. nitroreducens' encodes 38 genes for putative MHCs, with 8 being encoded within menaquinone (MK): cytochrome $c$ oxidoreductases gene clusters (Supplementary Data 6). A recent study revealed that 5 MK: cytochrome $c$ oxidoreductase gene clusters are conserved across the *Methanoperedenaceae* family, 4 of which are present in the '$Ca$. M. nitroreducens' genome. These clusters were hypothesized to mediate transfer of electrons from the menaquinone pool to MHCs outside the cytoplasmic membrane[18,50]. In the current study, significant differential expression of three MK: cytochrome $c$ oxidoreductase gene clusters was observed when '$Ca$. M. nitroreducens' conducted extracellular respiration with different electron acceptors (Fig. 4). The Group 5 MK: cytochrome $c$ oxidoreductases (Methanoperedens_01167 and 01168) showed the highest expression when '$Ca$. M. nitroreducens' utilized nitrate as a terminal electron acceptor (standard redox potential + 433 mV). However, when respiring on the poised electrode (+400 mV), the Group 3 MK: cytochrome $c$ oxidoreductase (Methanoperedens_02529) was expressed 2-fold greater than in the other two conditions. In contrast, 4 MHCs within the Group 1 MK: cytochrome $c$ oxidoreductase cluster (Methanoperedens_02260, 02261, 02264, 02265) were at the highest levels of expression when '$Ca$. M. nitroreducens' used soluble iron (+370 mV) as the electron acceptor. Similar phenomena have been observed in electroactive bacteria, for example, *G. sulfurreducens* was shown to show differential expression of multiple quinol: cytochrome $c$ systems (ImcH and CbcL) dependent on the redox potential of electron acceptors[51,52]. The differential expression of the various MK: cytochrome $c$ oxidoreductase clusters observed in this study provides evidence for the ability of '$Ca$. M. nitroreducens' to adapt to changes of redox potentials of different electron acceptors.

Consistent with localization observation and electrochemical activity characterization of $c$-type cytochromes (Figs. 2a, b; 3a–d), MHCs predicted to be extracellular were also found to be highly expressed by '$Ca$. M. nitroreducens' (Supplementary Data 2, 6). Given '$Ca$. M. nitroreducens' couples dissimilatory nitrate reduction using a *Nar* reductase (Fig. 1f), its high expression of MHCs when respiring with nitrate is interesting. Of these MHCs, 5 predicted to be extracellular were highly expressed, including the two most highly expressed (5-heme, Methanoperedens_00391; 8-heme, Methanoperedens_00559) (Supplementary Data 5). The high expression of extracellular MHCs for nitrate-fed '$Ca$. M. nitroreducens' was consistent with the positive staining of surface-associated extracellular cytochromes (Supplementary Fig. 3a, b), indicating that this species may use an alternative MHC-based EET pathway to achieve nitrate reduction via DIET with co-abundant denitrifying populations (as suggested

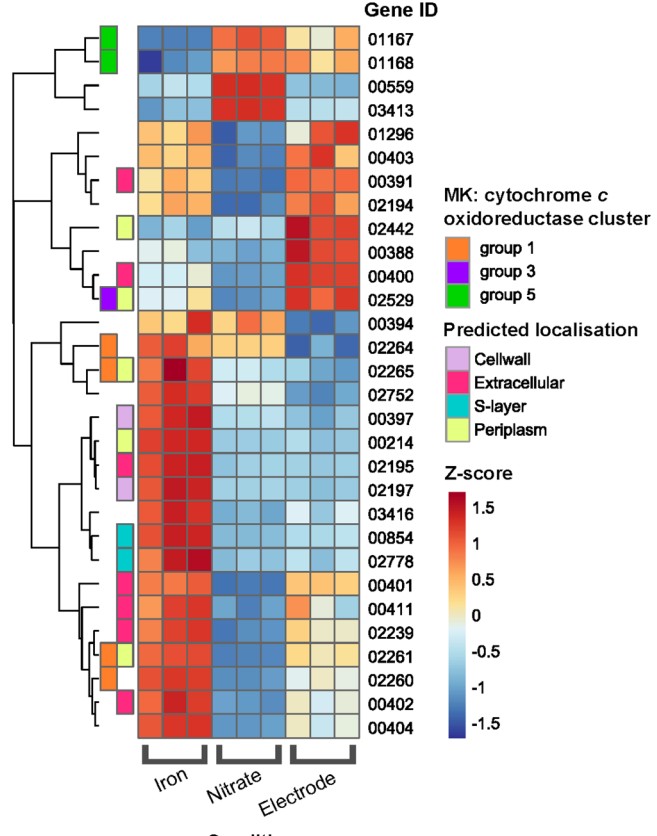

**Fig. 4 | Differentially expressed ($p$-adjusted < 0.05) putative multiheme $c$-type cytochromes (MHCs).** $P$-values were calculated via the R Package DESeq2, in which the Wald Test followed by the Benjamini and Hochberg method for adjusted $p$-values was applied (Supplementary Data 2). The three biological replicates for each batch condition are displayed side-by-side. MHCs were predicted by identifying proteins with ≥ 3 CXXCH amino acid motifs and localisation was predicted using PSORTb v.3.0.3[91], as well as identifying S-layer protein domains via the NCBI CD-Search Tool. 'MK: cytochrome oxidoreductase cluster' refers to proteins that fall within a menaquinone: cytochrome $c$ oxidoreductase group defined by Leu et al.[50].

previously[53]). When the electron acceptor was shifted from nitrate to soluble iron, 5 MHCs predicted to be extracellular were significantly upregulated (more than twofold change), out of which an extracellular 16-heme MHC (Methanoperedens_00214) and an extracellular 8-heme MHC (Methanoperedens_02195) saw the highest upregulation of 38.2 and 7.6-fold, respectively. The 8-heme MHC (Methanoperedens_02195) has homologous protein domains to OmcZ; a loosely bound outer surface MHC which has been well-characterised amongst *Geobacter* species. For *G. sulfurreducens*, OmcZ was found to be essential for long-distance extracellular electron transfer to electrodes[54,55], through the polymerization into highly conductive nanowires[56–58], and−like the OmcZ-like MHC of '$Ca$. M. nitroreducens'− was upregulated in the presence of soluble iron, suggesting similar roles in EET for these MHC homologs[59]. Other '$Ca$. M. nitroreducens' MHCs were also differentially upregulated when respiring on solid electrodes *versus* nitrate (Fig.4; Supplementary Data 2). Eight MHCs showed significant upregulation, including one 5-heme MHC (Methanoperedens_00388; 44.2-fold), a 5-heme extracellular MHC (Methanoperedens_00391; 6.7-fold) and a 12-heme extracellular MHC (4.4-fold; Supplementary Data 2). The distinct expression of extracellular MHCs observed here is similar to the metabolic characteristic of a model EET-capable microorganism, *Geobacter*, which was also identified to use different outer membrane conduits to respire a wide array of extracellular substrates[60]. Collectively, these MHC expression

patterns suggest that '*Ca*. M. nitroreducens' can use different MHC-based pathways to allow the use of different electron acceptors.

Taken together, we present a combination of novel methods to resolve the EET physiology and mechanisms of the ANME '*Ca*. M. nitroreducens'. The physiological evidence from fluorescence and single-cell electron microscopic visualization combined revealed the direct EET pathways of '*Ca*. M. nitroreducens'. Heme-reactive staining verified the localization of extracellular cytochromes and in vivo electrochemical characterization combined with chemically marking hemes with AcMet identified surface cytochromes as the conduits of '*Ca*. M. nitroreducens' for EET. Further spectroscopic analysis verified the active role of MHCs in mediating the electron transfer of '*Ca*. M. nitroreducens'. Though this EET pathway has been speculated to be a unifying evolutionary feature of ANME lineages, which generally have a high abundance of genes encoding MHCs[18,50], this study physiologically confirmed the pathway in ANME *Methanoperedenaceae*. Another intriguing finding of this study is that EET pathways of '*Ca*. M. nitroreducens' can be functionally altered by tuning expression levels of various MHCs to accommodate terminal electron transfer to substrates of varying redox potentials. Differential transcriptomic analysis of MHCs reveals that '*Ca*. M. nitroreducens' use different complexes of menaquinone cytochrome type *c* oxidoreductases as well as extracellular MHC conduits, respectively, to likely facilitate the electron transfer out of the menaquinone pool and to different terminal electron acceptors. This electron transfer characteristic resembles what has been described for the EET-capable model microorganism of *Geobacter*, which uses at least two MHC-based complex systems for quinol: cytochrome *c* oxidoreduction step[51,52], and multiple putative outer membrane MHC conduits for the final electron step to different electron acceptors[60]. Analogous to the general metabolic versatility of *Geobacter*[61], the tuneable EET pathways may provide ANME with the more metabolic capability than we currently know, which is likely achieved by the acquisition of MHCs via a hypothetic lateral gene transfer process[50]. Expressing and tuning an array of MHC-based electron transfer pathways could make ANME competitive under diverse fluctuating electron acceptor conditions for niche-specific adaptation. This study takes an important step in our knowledge of the EET pathways of ANME and contributes to our broader understanding of their key roles in linking the global methane cycle with other biogeochemical cycles.

## Methods

### '*Ca*. M. nitroreducens' enrichment
'*Ca*. M. nitroreducens' was collected from a 6.1 L parent sequencing batch reactor (SBR) for inoculation and characterization. The parent SBR was fed with methane, nitrate and ammonium to enrich '*Ca*. M. nitroreducens' that performs nitrate-dependent AOM. '*Ca*. M. nitroreducens' dominated the enrichment consortium at a stable abundance of ~30–35%. Detailed operational conditions of the SBR and the synthetic medium composition were described previously[4].

### Incubation of '*Ca*. M. nitroreducens' enrichment for respiratory iron reduction
To verify the EET capability and distinguish the direct and indirect EET pathways of '*Ca*. M. nitroreducens', the enrichment culture was first incubated to test its ability to perform iron-dependent AOM. Incubations were performed in 4.5 mL Exetainer vials (Labco, U.K.) with sacrificial sampling. In the first step, biomass (with a protein concentration of $56.4 \pm 4.3$ mg/L) collected from the parent SBR was transferred to an anoxic bioreactor vessel (210 mL) and cultivated with methane feeding until the residual nitrate depleted. The nitrate-free biomass was flushed with a gas mix $Ar:CO_2$ (95:5, vol:vol) for 15 min to strip the residual dissolved methane. The prepared biomass was then split into two treatment groups. The first group was amended with ferric citrate from a 50 mM stock solution to

serve as the terminal electron acceptor, giving an iron concentration of 5 mM. Another group, as the control, was amended with nitrate as the terminal electron acceptor at a final concentration of 5 mM. Ammonium was also amended in this group at 5 mM to scavenge the nitrite (from nitrate reduction) by anammox bacteria also present. Aliquots of biomass (2 mL) from both treatment groups were then inoculated into Exetainer vials for incubation. The vials were cap-sealed and 1.5 mL of $^{13}C\text{-}CH_4$ (>99.99% 13 C; Sigma-Aldrich) was injected into the headspace using a gas-tight syringe (81520, Hamilton). All these steps were performed inside an anaerobic chamber (Coy Laboratory Products Inc., USA). The vials were incubated at room temperature ($24 \pm 2$ °C) on a shaker at 120 rpm. Sacrificial samplings were carried out throughout the 54 h incubation to monitor the correlation between methane oxidation and iron/nitrate reduction. Specifically, three vials were harvested at each sampling time (5, 14, 28, 46, and 56 h after the initial setup) and acidified with anaerobically prepared HCl stock solution (1 M) to extract dissolved $CO_2$. Vials were then kept for at least 1 h for headspace equilibration before the gas phase was sampled and analysed for $^{13}CO_2$ with a gas chromatography-mass spectrometry (7890A/5975C; Agilent). Gas chromatography (GC) was performed using a J&W HP-PLOT Q PT column (30 m × 530 μm; Agilent, United States) with Helium used as carrier gas at a constant gas flow of 5.58 ml/min. The GC oven temperature was held at 50 °C for 2 min with an injection volume of 250 μL in split mode with a split ratio of 25:1. $CO_2$ was detected in positive electron impact (EI) at 70 eV using the standard autotune procedure for mass calibration. Acquisition was performed in Total Ion Chromatography (TIC) for identification and in Selected Ion Monitoring (SIM) for quantitation purposes monitoring m/z signals at 44 and 45 Da with a dwell time of 30 ms for each signal. Data processing was performed using the Chemstation program (Agilent, United States). Production of ferrous iron over time in the liquid phase was measured using a spectrometric ferrozine methodology. Nitrate and ammonium were measured with a Lachat QuickChem8000 Flow Injection Analyzer (Lachat Instrument, Milwaukee, WI)

### Discrimination of iron reduction pathways of '*Ca*. M. nitroreducens' by fluorescent visualization
'*Ca*. M. nitroreducens' in the enrichment culture could potentially perform iron-dependent AOM independently or with interdependent iron reducers, via a direct non-syntrophic or an indirect syntrophic EET pathway[21]. To discriminate these two pathways, a fluorescent method was established utilising $Fe^{2+}$-selective chemosensory fluorescence and autofluorescence of '*Ca*. M. nitroreducens'. The general principle of this method is as follows. Ferrous ions produced from microbial ferric reduction could accumulate on cell surfaces via absorption to redox proteins. Thus the iron-reducing microbes could be visualized with $Fe^{2+}$ chemosensory fluorescence[24]. Meanwhile, '*Ca*. M. nitroreducens' cells can be visualized via the autofluorescence of cofactor $F_{420}$[10].

Specifically, biomass samples were collected at the end (at 54 h) of the same isotopic incubation as noted above and washed twice with fresh medium. Rinsed cells were then stained with 25 μM $Fe^{2+}$-specific probe (FeRhoNoxTM; SCT037; Merck) at 37 °C for one hour, washed again, and resuspended in the fresh medium. All procedures were carried out inside the anaerobic chamber. The fluorescence imaging was carried out using a confocal scanning laser microscopy (CLSM; LSM 900, ZEISS). Autofluorescence was visualized with track channel at an excitation light of 405 nm and an emission wavelength of 465 nm, while FeRhoNoxTM fluorescence was with track channel at a laser excitation of 532 nm and an emission wavelength of 570 nm. To verify the fluorescent specification of the $Fe^{2+}$ chemosensory fluorescence and autofluorescence at different emission wavelengths, two additional fluorescent incubation controls were carried out: (1) The '*Ca*. M.

nitroreducens' enrichment culture was mixed with 5 mM $Fe^{3+}$ and immediately stained (i.e. without the iron-reducing incubation process) with 25 μM FeRhoNoxTM according to the above-described procedures; (2) The 'Ca. M. nitroreducens' enrichment culture collected at the end of iron-reducing incubation and visualized without FeRhoNoxTM staining. To examine whether 'Ca. M. nitroreducens' was the direct iron reducer in the consortium, the overlapping ratio of cells stained with FeRhoNoxTM and cells with autofluorescence was quantified based on the congruency of FeRhoNoxTM chemosensory fluorescence signals and autofluorescence signals. Quantification was performed with the *daime* software, according to procedures described by Daims[62]. Binary thresholds for each fluorescent image were independently selected to distinguish FeRhoNoxTM-stained or autofluorescence cells from background fluorescence. The congruency was calculated as the average threshold pixel count in a given channel across 20 fields of view averaged across the corresponding values.

## Visualization of reductive silver deposition on cell surface of 'Ca. M. nitroreducens'

To test whether 'Ca. M. nitroreducens' can use surface-localized cytochromes to mediate metal reduction, the 'Ca. M. nitroreducens' enrichment culture was tested for its ability to reduce silver, with the potential deposition of silver particles on the cell surface of 'Ca. M. nitroreducens' visualized with electron microscopy. Specifically, aliquots of 2 mL nitrate-free 'Ca. M. nitroreducens' enrichment culture was inoculated into 4.5 Exetainer vials. An $AgNO_3$ stock solution was supplied to reach a final concentration of Ag (I) at 200 μM or 500 μM. Sealed vials were injected with 1 mL of methane in the headspace to supply electron donor. To establish cytochromes as the active silver reductase, a parallel incubation with 200 μM Ag(I) supplement was performed in the presence of 100 mM of acetyl methionine (AcMet), which was previously shown to deactivate *c*-type cytochromes through heme ligation[30,38]. All inoculation steps were prepared inside an anaerobic chamber and incubation carried out in the dark at room temperature ($24 \pm 2$ °C) on a shaker at 120 rpm. Silver reduction by the 'Ca. M. nitroreducens' enrichment culture was expected to be accompanied by a colour change of the suspension. Once a colour change was observed (12 h incubation), biomass from incubation was collected to have single-cell visualization of silver deposition on 'Ca. M. nitroreducens' cells using sectioning transmission electron microscopy (TEM).

## Transmission electron microscopy

Biomass samples were taken at the end of the iron- and silver-reduction experiments and directly from the parent reactor, and pelleted by centrifugation at $5000 \times g$ for 5 min prior to immediate fixation in 2.5% glutaraldehyde (in 50 mM HEPES; pH 7.4) overnight at 4 °C. The fixed samples were washed three times in 50 mM HEPES, pH 7.4, 35 g/L NaCl, with sequential resuspension and centrifugation (2 min, $4000 \times g$). Biomass samples were then added with a similar amount of melted agarose. After spinning the mixed solution at $6000 \times g$ for 30 s, samples were kept on a cooling block for 10 min. The bottom of the tubes was cut with a dog toenail cutter and samples embedded in agarose blocks were plugged out using wooden sticks. Samples were plugged on Menzel glass, covered in HEPES buffer and cut into little 1-micron sections ($\sim$1 mm³). To visualize the localization of active cytochromes on 'Ca. M. nitroreducens', cytochrome reactive staining was then applied. Specifically, section samples were subjected to heme-active staining with 3,3'-diaminobenzidine tetrahydrochloride (DAB). An $H_2O_2$/DAB solution at $H_2O_2$ final concentration of 0.02% and DAB final concentration of 0.0015 g DAB/ml buffer (50 mM Tris HCl pH 8) was prepared according to details described by McGlynn et al.[13]. This solution was added to embedded sections for staining for 2 h at room temp. A DAB solution without $H_2O_2$ was added to a separate set of samples for comparison.

After staining, the DAB solution was removed with $5 \times 2.5$ mL washes with 50 mM HEPES, pH 7.4, 35 g/L NaCl. Washed sections were then covered with 1% $OsO_4$ working solution and this was incubated inside a microwave (Biowave® 34700 Microwave with SteadyTemp™ Cooler, Ted Pella Inc., Redding, CA, USA) for 6 min. The Biowave was set at 150 W with a 74.5 kPa vacuum on during the incubation, and samples were incubated with a program (2 min on, 2 min off and then 2 min on) repeated twice. The samples were then washed with $5 \times 2.5$ mL washes with 50 mM HEPES, pH 7.4, 35 g/L NaCl. Washed samples were then dehydrated with a graded solvent (acetone) series (50%, 70%, 90%, 100% × 2 times). For each dehydration step, samples were incubated inside the Biowave (250 W and 74.5 kPa vacuum on) for 40 s. The dehydrated samples were then treated with resin infiltration at graded resin series (resin in acetone at 33%, 50%, 66.6%, 100% × 2 times). For each infiltration step, samples were incubated inside the Biowave (at 250 W and 74.5 kPa vacuum on) for 3 min. Samples were then moved to a silicon form that had been filled with 100% resin followed by a placement in a 60 °C oven overnight for resin polymermization.

Thin (70 nm) sections with embedded samples were then collected on copper microgrids for TEM visualization. The fixed samples from the silver-reducing incubation were prepared with the same procedures but without heme staining. The obtained resin blocks were sectioned either at a thickness of 70 nm for TEM visualization, or at 150 nm for energy-dispersive X-ray spectroscopy (EDS) analysis, and floating sections were also mounted on copper grids. TEM was performed at 80 kV using a Hitachi HT7700 microscope. EDS was analyzed on HT7700A coupled with a Bruker QUANTAX EDS system. Single 'Ca. M. nitroreducens' cells were distinguished in TEM images according to their unique morphology, based on extensive fluorescence in situ hybridization (FISH) characterization of the consortium: (1) they normally grow with irregular cocci shape at larger size to other populations, and they are typically found as sarcina-like clusters[4]; (2) they are the only population in the consortium with detectable polyhydroxyalkanoate (PHA) granule storage[53], which can also be observed with TEM.

## Whole-cell electrochemical characterization

Electrochemical characterization of the 'Ca. M. nitroreducens' enrichment culture was established using single-chamber, three-electrode electrochemical cells inside an anaerobic chamber. The electrochemical cells (Supplementary Fig. 6) were manufactured with 3D printing. The working electrode (WE) was 1.0 mm thick glass substrate (2 cm × 2 cm) coated with fluorine-doped tin oxide (FTO) (Xop Fisica, Spain). Four FTO substrates were installed on the side ports of the electrochemical cell, sealed with silicon glue. A platinum wire was used as the counter electrode (C.E.), and an Ag/AgCl electrode (RE-5B, BASi, USA) as the reference electrode (R.E.). The inner active volume of the electrochemical cell was 15 mL.

10 mL of the 'Ca. M. nitroreducens' enrichment culture collected from parent reactor was used to inoculate the electrochemical cells. The prepared biomass was resuspended in 10 mL of high-conductivity medium and inoculated into the electrochemical cell with headspace filled with a mixed gas ($CH_4$:$CO_2$:$N_2$ = 90:5:5). The cell suspension was gently mixed using a magnetic stirrer at 100 rpm.

Chronoamperometry (CA) and cyclic voltammetry (CV) were used to characterise the electrode reduction by the 'Ca. M. nitroreducens' enrichment culture. CA was conducted at a set potential (0.4, 0 and −0.2 V vs. SHE) to depict the correlation between polarized potential and current production. Whole-cell CVs were profiled at different time intervals (2, 8, 16, 24 h) for the electrochemically cultivated 'Ca. M. nitroreducens' enrichment culture at the potentiostatic condition (at 0.4 V vs. SHE). To distinguish the contact-based and diffusion-dependent redox response of the 'Ca. M. nitroreducens' enrichment culture, after 24 h of electrochemical cultivation cell suspensions were

filtered through 0.22 μm membrane to remove cells. The filtered solution was electrochemically tested with CV in a newly established electrochemical cell with fresh medium as a counterpart. All CVs were recorded at a scan rate of 10 mV s⁻¹ and a scan range from −0.5 to 0.4 V vs. SHE. All electrochemical characterizations were analysed using a potentiastat (VMP3, Biologic Science Instrument, France).

To further identify the role of c-type cytochromes in EET by the 'Ca. M. nitroreducens' enrichment culture for electrode reduction, the electrochemical behaviour was examined with treatment of AcMet. In a parallel electrochemical cell where 'Ca. M. nitroreducens' enrichment culture was electrochemically cultivated (at 0.4 V vs. SHE), AcMet was supplemented to a concentration of 100 mM at 24 h. The chronoamperometric current response was recorded over time. Meanwhile, whole-cell CVs were also profiled in the presence of AcMet.

## Operation of bioelectrochemical systems for electrode-dependent AOM

BESs were set up to investigate the process characteristics and metatranscriptomic dynamics when 'Ca. M. nitroreducens' actively respiring on electrodes. The configuration of each BES (Supplementary Fig. 9) and parameter details were described in our previous study[33]. Carbon fibre (AvCarb C100, Fuel Cell Store, USA) was used as the anodes. It was pre-treated with $N_2$ plasma to render the hydrophilic surface[63], and then washed with acetone and RO water each for three times. The fibre was cut into 1 cm × 1 cm square pieces, four of which were subsequently skewered on a titanium wire (diam. 0.5 mm, Sigma, USA), to produce 'fibre-skewer' electrodes. Five processed electrodes were evenly located into the anode chamber (Supplementary Fig. 9). Platinum mesh was used as the cathode and Ag/AgCl as a reference electrode (RE-5B, BASi, USA).

For inoculation, 60 mL of the 'Ca. M. nitroreducens' biomass was collected from the parent reactor and centrifuged at $5000 \times g$ for 8 min. The biomass pellet derived was then washed three times inside an anaerobic chamber (Coy Laboratory Products Inc., USA) with a nitrate- and oxygen-free high-conductivity medium as described previously[33]. The 'Ca. M. nitroreducens' enrichment culture was finally resuspended in 160 mL fresh medium and then inoculated into the anode chamber of the BES. Sterile 50 mM phosphate buffered saline (PBS) of 160 mL was used as cathodic electrolyte. Following replacement of 30 mL of headspace gas in the anode chamber with ¹³C-methane, two BESs were operated in batch mode inside an anaerobic chamber (22 ± 2 °C) as duplicates. The anode chambers were mixed with magnetic stirrers at 200 rpm. The anode was poised at 0.4 V versus SHE with a potentiostat (CHI 1030C, CH Instruments Inc, USA), which also recorded the chronoamperometric current. 0.5 mL of liquid sample was periodically (on day 3, day 6, day 13, day 20, day 27, day 37, day 44, day 51, day 61, day 72) withdrawn from the anode chamber over 72 days using a 3 mL sterile syringe with 25-gauge needle. The sample was immediately injected into a 3 mL, helium-flushed Exetainer vials (Labco), which was preserved at −20 °C for later measurement of ¹³$CO_2$. For the quantification of ¹³$CO_2$ production from ¹³$CH_4$, 0.2 mL 1 M HCl was injected into each thawed sample to extract $CO_2$ prior to measurement. The extracted $CO_2$ in the vial headspace was analysed with a gas chromatography-mass spectrometry (7890A/5975C; Agilent) for the ¹³C isotope ratio, based on which ¹³$CO_2$ production was calculated as described in Scheller et al.[8].

Cumulative charge (the amount of electrons produced) was calculated by integrating the current profile logged by the potentiostat. Current efficiencies of AOM were represented by the comparison between cumulative charge and electrons theoretically produced from the measured conversion (8 e⁻ for ¹³$CO_2$ from ¹³$CH_4$).

## Fluorescence in situ hybridization (FISH)

Spatial distribution of 'Ca. M. nitroreducens' cells in the biofilm formed on the surface of the carbon fibre electrodes was examined with FISH after 72 days of operation of BESs. Square fibre pieces were collected from titanium wires using a sterilised tweezer and laid them on the bottom of a 50 mL specimen jar, which was stored on ice. After moving samples out of the anaerobic chamber, 2 drops of 2% agarose were gently added to the surface of the electrode and allowed to set. For fixation, 4% paraformaldehyde solution was gently added to just submerge electrodes. The samples were then kept at 4 °C for 4 h. Paraformaldehyde -fixed samples were then washed with 1× PBS for three times. 30% sucrose solution (filter sterilise before use) was then added to completely submerge electrode samples, which were then kept at room temperature for ~1 h. Afterwards, each electrode sample was vertically placed inside a cubic plastic mould (ca. 2 × 2 × 2 cm), which was then filled with OCT and placed at −80 °C in a sealed sample bag. Sections were then cut from these electrode-OCT preparations with a Cryostat NX70 (Leica Biosystems) to a thickness of 20 μm and immobilised on gelatin-coated (1.25% w/v) glass slides, air dried and stored at −80 °C. Fluorescence in situ hybridisation was performed essentially as described by Daims and colleagues[64]. Microscopic analysis was performed with a LSM710 (Carl Zeiss) laser scanning confocal microscope. The EUBmix (5′-GCTGCCTCCCGTAGGAGT-3′; 5′-GCAGCCACCCGTAGGTGT-3′; 5′-GCTGCCACC CGTAGGTGT-3′)[65,66], Geo3-A (5′-CCGCAACACCTAGTACTCATC-3′)[67] and AAA-641(5′-GGTCCCAAGCCT ACCAGT-3′)[68] probe sets were used to target bacteria, *Geobacter* and 'Ca. M. nitroreducens', respectively. The NON-EUB probe was used as a negative control for non-specific probe binding[69].

## Scanning electron microscopy (SEM)

Biofilm-colonized square electrodes were collected from BESs after 72 days of operation using sterilised tweezers and immediately submerged into 2.5% glutaraldehyde (in 50 mM HEPES; pH 7.4) and stored at 4 °C. A standard microwave-assisted dehydration methodology was used for sample preparation as described elsewhere[70]. Dried samples were mounted onto aluminium stubs using conductive double-sided adhesive carbon tabs (ProSciTech). The stubs were then sputter-coated with platinum (EIKO IB-5 Sputter Coater, EIKO Engineering Co. Ltd., Hitachinaka, Japan) to achieve a uniform layer over the membrane surface. SEM images were obtained using a JOEL-7100F Field Emission Scanning Electron Microscopy (FESEM).

## In situ spectroelectrochemical characterization with Resonance Raman spectroscopy

To further confirm that c-type cytochromes mediate the EET of 'Ca. M. nitroreducens' for electrode reduction, we conducted in situ spectroscopic characterization of electroactive biofilm of the 'Ca. M. nitroreducens' enrichment culture. A single chamber spectroelectrochemical cell manufactured by 3D printing (Supplementary Fig. 14b), was established for both electrochemical and confocal Raman measurements. It allowed for simultaneous characterization of electrochemical and spectroscopic properties of 'Ca. M. nitroreducens' in vivo and in situ.

To achieve in situ spectroelectrochemical characterization, biofilms of 'Ca. M. nitroreducens' enrichment culture were firstly cultivated on the graphite working electrodes (8 mm × 8 mm × 2 mm) (Graphite Sales Inc., U.S). Our previous study indicated that other electroactive microorganisms that harbour c-type cytochromes could grow by utilizing organics released from biomass during long-term electrochemical cultivation of 'Ca. M. nitroreducens'. In the present study, we established a 'Ca. M. nitroreducens'-dominant biofilm on the disconnected graphite electrodes with nitrate as the only electron acceptor to exclude the potential growth of other undesirable EET-active microorganisms. Specifically, multiple graphite electrodes were placed into a 60 mL incubation serum bottle (Supplementary Fig. 14a). The bottle was inoculated with 40 mL 'Ca. M. nitroreducens'

 

enrichment culture collected from the parent SBR. Methane, nitrate and ammonium were supplied to the incubation to enable the growth of 'Ca. M. nitroreducens' in suspension and on the surface of the submerged graphite electrodes. Formation of 'Ca. M. nitroreducens' biofilm on the electrode surface after 128 days of incubation was visualized by autofluorescence imaging of 'Ca. M. nitroreducens' cells according to methods described in a previous section.

Biofilm-covered graphite electrodes were removed from the serum bottle and were rinsed in the degassed sterile fresh medium to remove residual nitrate. The electrode was then assembled immediately into the spectroelectrochemical cell, with the biofilm facing the coverslip window at ~1 mm distance (Supplementary Fig. 14b). 10 mL of methane-saturated medium was filled into the spectroelectrochemical cell and 'Ca. M. nitroreducens'-dominant biofilm was electrochemically incubated at +0.4 V vs. SHE for 24 h with methane supplemented in the headspace. Right before the Raman measurements, the medium in the spectroelectrochemical cell was replaced and topped up with methane-free medium. All these operations were carried out inside an anaerobic chamber.

The electrochemical manipulation and spectroscopic Raman measurements were conducted in a purposely built unit (Supplementary Fig. 14c). Time-resolved simultaneous Raman and CA measurements were carried out by continuous collection of Raman spectra from the 'Ca. M. nitroreducens' biofilm on the graphite electrode that was polarized at different potentials. Two constant potentials, i.e., +0.4 V and −0.3 V vs. SHE, were controlled to achieve different redox-states of 'Ca. M. nitroreducens' biofilm. Each potential was maintained for at least 15 min before performing Raman measurement to allow for stabilisation of the redox status of the 'Ca. M. nitroreducens' biofilm, which was confirmed by stabilized chronoamperometric current. The spectroelectrochemical cell was physically moved to collect Raman spectra at both potentials from different spots of the biofilm.

The Raman measurements were performed using an Alpha 300 Raman/AFM (WITek GmbH, Ulm, Germany) equipped with an air objective lens (Nikon 40×, N.A. 0.6, CFIS Plan Fluor ELWD objective). A frequency-doubled continuous-wave Nd:YAG laser at 532 nm was used for excitation. The collar ring was fixed to 0.16 mark for collar correction. Raman signals were collected with a 50 mm optical fibre with a resolution of 4 cm$^{-1}$. The spectra were acquired at an integration time of 0.2 s to obtain high-contrast resonance spectra. The Project FOUR software (WITec GmbH, Ulm, Germany) was used to process spectra. The OriginPro 9.1 software (OriginLab, Northampton, USA) was used for data fitting. Raman spectra reported for each potential are the means of the quality-filtered band intensities obtained from multiple spots (n > 10) through 4 independent electrode-biofilm samples.

## Metagenomics and metatranscriptomics analysis

**RNA and DNA extraction.** Metagenomic and metatranscriptomic analyses were performed for biomass samples collected from iron-reducing incubation batches and BESs. Three vials were harvested (as three independent biological replicates) at 24 h in the iron respiratory reduction batch from both the iron and nitrate treatment, respectively. 1 mL of biomass from each collected vial was immediately sampled into the lysis buffer of the FastRNA Pro Soil-Direct Kit (MP Biomedicals, USA) and RNA was then extracted following the manufacturer's protocol. The remaining 1 mL of biomass was preserved at −80 °C for DNA extraction using a FastDNA SPIN kit for soil (MP Biomedicals, USA) following the manufacturer's protocol.

Samples for BESs were taken from electrode's biofilm of two reactors operated in parallel after 72 days of operation. Five pieces of square carbon fibre electrodes were collected from different titanium wires using sterilized tweezers as representative biofilm samples. Triplicate biofilm samples were collected in the anaerobic chamber for an immediate co-extraction of RNA and DNA with the RNeasy PowerSoil Total RNA Kit (QiAGEN) according to the manufacturer's instructions. 10 mL of supernatant was also withdrawn from each BES for RNA and DNA extraction. The DNA and RNA concentrations of all samples were quantified using the Quant-iT dsDNA and dsRNA HS assay kit (Invitrogen). 5% (vol/vol%) RNAseOUT (Invitrogen) was added in the extracted RNA sample before preservation at −80 °C. Prior to sequencing, DNA was further removed from RNA extracts with RNA Clean & Concentrator-5 kit (Zymo Research) with an in-column DNase I treatment step (repeated twice).

**DNA and RNA library preparation.** Paired-end libraries were prepared for all DNA extracts using the Illumina DNA Prep Kit (Illumina). The libraries were quantified using the Qubit dsDNA HS assay kit (Invitrogen) and sequenced using the Illumina NovaSeq to produce at least 5 Gb of shotgun metagenomic paired-end reads (2 × 150 bp). Prior to library prep, RNA integrity was assessed using a TapeStation (Agilent). Ribosomal-RNA depletion, cDNA generation and library preparation were done using the Illumina Ribozero Plus Kit following manufacturers protocol by the Australian Centre for Ecogenomics (ACE). RNA libraries were sequenced on the Novoseq6000 by ACE, generating at least 3 Gb per 30 million paired-end reads (2 × 150 bp) per sample.

**Sequence data assembly and recovery of population genomes.** Paired-end reads for the metagenome were quality trimmed using cutadapt v4.2.0[71] and assembled into metagenomes using metaSPAdes v3.11[72], both with default settings. MAGs were then recovered from the assemblies using the default binning pipeline embedded in Aviary v0.5.0 (https://github.com/rhysnewell/aviary) which utilises Minimap2[73] and CoverM v0.6.1 (https://github.com/wwood/CoverM) for read mapping, multiple binning algorithms[74–78], followed by use of DASTool[79] and CheckM[80] for bin refinement. Completeness and contamination of the final bins outputted by Aviary was assessed using CheckM2[81], retaining MAGs with ≥70% completeness and ≤10% contamination (besides 'Ardenticatenaceae-3', which returned 11.6% contamination but was retained due to its estimated 100% completeness). These MAGs were pooled with near-complete MAGs derived from a previous study[53], which were generated from the same parent bioreactor as the current study, and redundant MAGs were dereplicated using the CoverM v0.6.1 Cluster function with default settings (ANI > 99%). MAGs were classified with GTDB taxonomy (release 07-RS207) using the GTDB-Tk v2.1.0 Classify function[82]. Relative abundance community profiles of each metagenome were determined using CoverM v0.6.1 Genome function with the dereplicated MAGs.

**Functional annotation.** Functional annotation of the microbial community was done using METABOLIC-G v4.0[83], which first calls open reading frames (ORFs) using Prodigal[84], and performs a hmmsearch[85] against three HMM databases derived from KOfam[86], TIGRfam[87], and Pfam[88]. Functional annotation of 'Ca. M. nitroreducens' was further curated manually by consulting published annotations of the complete genome[53]. Putative MHCs were predicted across all MAGs by identifying ORFs with 3 or more CXXCH amino acid motifs, and confirmed by searching for conserved cytochrome c-type protein domains via NCBI Batch CD-Search[89]. Putative MHCs were subsequently searched for signal peptide sequences using SignalP v6.0[90], and localization was predicted using pSORTb v3.0.3[91]. 'Ca. M. nitroreducens'-encoded MHCs that fall within a menaquinone: cytochrome c oxidoreductase cluster were annotated according to pre-identified homologous ORFs described in Leu et al.[7,92].

**Differential gene expression analysis.** The metatranscriptomic reads were first quality assessed using Kneaddata v0.10.0 (https://github.com/biobakery/kneaddata), and then trimmed using

 

Trimmomatic v0.39[93]. BAM files were generated by mapping trimmed reads to MAGs using CoverM v0.6.1 Make, and subsequently filtered via CoverM v0.6.1 Filter to exclude reads with <75% bases aligned and <95% overall identity. DNA contamination was assessed based on the directionality of transcripts and removed from BAM files using the Bioconductor R package, StrandCheckR v1.15.0. Finally, raw count tables were generated using the Subread v2.0.3 package (http://subread.sourceforge.net/), and RNA-transcripts per million (TPM) values were calculated using the following equation:

$$TPM = \frac{rg*rl*10^6}{flg*T} \qquad (1)$$

Where rg = reads mapped to gene g, rl = read length, flg = feature length (or CDS length), and $T$ = sum of rg*rl/flg for all genes. The percentage of unmapped reads for each metatranscriptome was determined using SAMtools[92] flagstat on the decontaminated BAM files. Differential gene expression analysis of 'Ca. M. nitroreducens' ORFs was done using the R package DESeq2 v3.15[94] with the raw count tables. A regularised log transformation was applied to the normalised gene expression count table generated by DESeq2. To identify differentially expressed genes, a Wald Test followed by the Benjamini and Hochberg method for adjusted $p$-values was applied (Supplementary Data 2). Genes were considered differentially expressed when the $p$-adjusted value was <0.05. The log-fold change in gene expression of differentially expressed putative MHCs across the nitrate, iron, and electrode conditions were visualised using the R-package pheatmap v1.0.12, with genes being clustered using the 'Euclidean distance' metric.

## Statistics and reproducibility
Unless indicated otherwise, all experiments were performed with $n = 3$ replicates (3 biological replicates). The long-term BES experiment was operated with $n = 2$ independent replicates. The Raman measurements were performed with $n = 4$ independent replicates. More details of reproducibility can be found in "Reporting Summary". To identify differentially expressed genes, a Wald Test followed by the Benjamini and Hochberg method for adjusted $p$-values was applied. Genes were considered differentially expressed when the $p$-adjusted value was <0.05.

## Reporting summary
Further information on research design is available in the Nature Portfolio Reporting Summary linked to this article.

## Data availability
All raw metagenomic and metatranscriptomic reads generated in this study have been deposited in NCBI database with BioProject ID PRJNA1003560, BioSamples SAMN36897499 to SAMN36897508 (https://www.ncbi.nlm.nih.gov/biosample?LinkName=bioproject_biosample&from_uid=1003560), and Sequence Read Archive IDs SRR25580091 to SRR25580106 (https://www.ncbi.nlm.nih.gov/sra?LinkName=bioproject_sra_all&from_uid=1003560). Source data and Supplementary Data are provided with this paper. Source data are provided with this paper.

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

## Acknowledgements

This work was supported by the Australian Research Council (ARC) through the Australian Laureate Fellowship (FL170100086; Z.Y.). S.M. and G.T. are supported by Australian Research Council (ARC) Future Fellowships FT190100211 and FT170100070, respectively. The authors are grateful to the ACWEB Analytical Services Laboratory (ASL) for all chemical analysis, T. Stark for the maintenance and operation of GC-MS, and the Centre for Microscopy and Microanalysis (CMM) at UQ for TEM/SEM analyses.

## Author contributions

X.Z. and S.H. conceived the study. X.Z., S.H and S.M. planned the experiments. X.Z. conducted the main experiments and analyses, including all batch incubations and corresponding chemical analyses, bioelectrochemical incubation and electrochemical characterization. X.Z., G.J. and S.M. performed the sampling, preservation, DNA and RNA extraction for metagenomics and metatranscriptomics sequencing. G.J. and A.L. performed the microbial community and bioinformatics analysis. J.Z. conducted the fluorescence microscopic visualization and processed fluorescent images. S.M. designed FISH probes and conducted FISH microscopy. X.Z. conducted the TEM microscopy and H.R. conducted the SEM microscopy. X.Z. and B.V. performed the Raman spectroelectrochemical experiment and B.V. analysed the Raman spectrum data. X.Z., S.H. and Z.Y. performed the process data analysis. X.Z., G.J., G.T., Z.Y., S.M. and S.H., wrote the manuscript in consultation with all other authors.

## Competing interests

The authors declare no competing interests.
