## [Peer Review File · Nature Communications]

REVIEWER COMMENTS

Reviewer #1 (Remarks to the Author):

In the manuscript 'Multi-heme cytochrome mediated extracellular electron transfer by the anaerobic methanotroph 'Candidatus Methanoperedens nitroreducens'' the authors describe several experiments that provide evidence that Candidatus Methanoperedens nitroreducens uses redox-active cytochromes for iron and electrode reduction. These experiments include differential transcriptomics, electrochemical techniques, Raman spectroscopy and microscopy.

Although I consider the topic of the manuscript interesting, I find that the experiments conducted, and the author's conclusions lack veracity and fail to reflect their intended findings. Furthermore, this manuscript lacks novelty as the literature already contains evidence, specifically referenced in the manuscript as reference 6, demonstrating that methanotrophic archaea utilize MHC for iron reduction.

In my opinion, the conclusions made are questionable due to their use of a culture stated by the authors as predominantly composed by *Ca. M. nitroreducens*, rather than a pure culture. This choice undermines the validity of their findings, as there is no evidence to confirm whether the observed results originate from the archaeon in question, or any other organism present in the culture. Indeed, it is described that only 38 and 32% of this species exists in the culture. While metatranscriptomic analysis may reveal the predominance of mRNA originating from a particular organism, it does not guarantee that all obtained results, such as those from electrochemistry and Raman spectroscopy, exclusively concern to that organism. This is clearly visible by the results obtained from the biofilm of the BES, where less than 10% of the mRNA was from *Ca. M. nitroreducens* and more than 23% was from *Geobacter*.

In addition to these issues, I also have additional concerns.

Line 92 – How the authors know if the characterized MHC are important for EET? Although they identify some of the MHC present in the genome of this organism, it does not mean that the cytochrome will be important for iron reduction.

Line 92-93 – How the authors ensure that both reactions (iron and nitrate reduction) are conducted by *Ca. M. nitroreducens*?

Line 109-110 – Depending on the speciation, iron citrate can enter the cells and be reduced in the periplasm space of Gram-negative bacteria. Since archaeon do not have outer-membrane but a cell envelope, there is no evidence that the reduction of iron citrate occurs at the cell surface. It may also occur at the inner-membrane. Furthermore, the reduction of Fe^{3+} will lead to the formation of Fe^{2+} that is soluble and will diffuse in the media.

Line 125-128 – it is not clear from the text how the MHC were quantified. A reference to the table of genes should be included.

Line 171 – It is not clear what acetyl methionine will do to cytochromes. The hemes in these proteins are typically hexacoordinated with two histidines, and there is no evidence in the literature that demonstrates that AcMet will have the capacity to replace the strong axial ligand Histidine. Considering that certain enzymes involved in the nitrogen cycle possess hemes with open coordination, it is plausible that AcMet could potentially interact with these cytochromes. However, it is important to note that such interactions would not impact electron transfer to iron or electrodes. This should be discussed in the manuscript.

Line 177 - TEM images – the authors describe that the nanoparticles are localized at the outer surface of *Ca. M. nitroreducens*, but from the figure it seems that they are at the EPS matrix and not at the cell surface. Given the figures observed with DAB, it seems that the cell surface is much more near the contours of the cell. Furthermore, it is not clear how the authors know that these are *Ca. M. nitroreducens* cells. This should be clear in the text.

Line 309-310 – If the authors are trying to identify MHC, they should know that all MHC have a signal peptide. While not all signal peptides undergo cleavage in these proteins, they are required to translocate the proteins outside the cytoplasm where the heme maturation process occurs. While in bacteria, the maturation occurs at the inner-membrane level towards the periplasmic space, in archaea this occurs towards the cell envelope.

Line 316-317 – The results obtained by the authors relative to the expression of the different cytochromes during different metabolic processes cannot be compared given that the number and type of other species in the cultures are different. The difference of the expression of the different proteins are not only due to the different metabolic processes but may also be influenced by the different species and compounds existent in the culture.

Reviewer #2 (Remarks to the Author):

The study of Zhang et al. characterizes the physiology of *Candidatus Methanoperedens nitroreducens* with state-of-the-art spectroscopic methods, electrochemical procedures and metatranscriptomics.

The experiments are difficult since only enrichment cultures of these archaea are available that predominantly grow in syntrophy with anammox bacteria.

General comments

The study is a collection of many different methods, whereby the focus lays on the role of the multi-heme c type cytochromes.

The experiments are well-conducted and involve custom-made bioelectric set ups to study methane oxidation coupled to the transfer of the electrons to electrodes.

The collection of data reported here should be published since they can be valuable to the specialist in the field of electron-conducting proteins and for the field of anaerobic methane oxidation in general.

While a good set of data has been acquired, no big fundamentally new discovery has been made, and it is difficult to draw a short, relevant conclusion from the study. My judgement of "collection of data" is substantiated by the fact that the authors write often "further support", "confirming", "further supporting", "further insight" etc. throughout the manuscript.

It could be considered to publish this work in a more specialized journal.

Selected comments by line number:

- line 34-37: This part should be elaborated more in the main manuscript, if this is the conclusion of the abstract

- line 73 "seldomly conducted": please cite where it has been conducted

- line 86 ff: even though the methods come later, it would be nice to know here already what we are talking about. That it is an enrichment culture coupled to ammonium oxidation (right?) that is switched to the (perhaps artificial) metabolism of metal reduction.

- line 89 "suggested": didn't they obtain a culture where this process really happens?

- line 111 "to better understand": in the conclusion it should be more clearly stated what is now clearer thanks to this study.

- line 131 (fig. 1): The quality of the figure is not sufficient. I cannot read panel F, for example since too small font anyway and it appears blurry. Panel B-D: where are the partner bacteria? Are those mostly archaea? I have missed a comment on that, or it is not explained.

- line 162 ff: How specific is this Ag⁺ method? It seems that Ag⁺ is quite reactive anyway. Should be discussed how indicative this is.

- line 164 "further support": this is throughout the paper. It is ok like this, but it highlights that the whole study is a collection of data that all point into this direction, but not a solid finding.

- line 236: This “AcMet” seems to appear without warning. I’m not aware of this method. Would be good to briefly discuss what this is and what it tells.

- line 349: I guess this is the “conclusion”, which highlights that there is not so much specific conclusion from this study (even though the study is valuable for the scientific community).

Reviewer #3 (Remarks to the Author):

Comments to Zhang et al.

The manuscript entitled “Multi-heme cytochrome mediated extracellular electron transfer by the anaerobic methanotroph ‘Candidatus Methanoperedens nitroreducens’” by Zhang et al. presents evidence from various visualization, electrochemical and spectroscopic characterization, and genomic analysis to support the hypothesis that extracellular multi-heme cytochromes (MHCs) play a critical role in the EET pathways of “Ca. M. nitroreducens”. The hypothesis was frequently invoked in both microbiology- and biogeochemistry-centered studies of metal-dependent ANME. However, evidence beyond genomic data were unfortunately lacking. This work therefore should fill an important knowledge gap in this regard. The experiments were well-designed, the collected data extensive and convincing, and the manuscript was well-written and precise. I have only a few minor comments to the authors at some places that need to pay attention to.

Minor comments

Lines 105-108: this sentence is confusing. What does the phrase “in the present of Fe³⁺ but without incubation” mean? Does it mean for the “Ca. M. nitroreducens” culture it was too soon after the Fe³⁺ addition to have activity? Also, it is unclear what “in the absence of staining” mean.

This manuscript reports some sequencing data of both metagenome and metatranscriptome. There are also some MAGs were generated in the study. These data should be publicly accessible in some way. Probably adding a separate data availability section can address this issue.

For Fig. 1F: I first can only see the blue bars, but not the others. I think it is important to clarify that this panel share the same figure key with Fig. 1E, if my understanding is correct.

Response to the reviewers' comments

We thank the reviewers for their valuable comments and the editor for offering the opportunity for manuscript revision. We hereby submit our responses and the revised manuscript for further review. Our responses to the comments have been divided and numbered to facilitate review. The reviewers' comments are in BLACK (12-point Italic font), our responses are in BLACK (12-point normal font), and the revised texts are in BLUE (11-point blue font).

Before the point-by-point responses, we would like to provide a general response regarding the novelty of this work:

The MHC-based EET pathway has been widely hypothesized for ANME in recent years. However, current knowledge on this is limited to metabolic prediction based on genomic and metatranscriptomic analyses. This is mainly because no pure culture of any ANMEs is available for biochemical characterization and genetic manipulation. Thus, empirical and physiological evidence is rarely reported, with only one case study in *Nature* providing single-cell visualisation-based evidence¹. Without *in vivo* identification and biophysical and biochemical characterization, the MHC-based EET pathways of ANME remain speculative².

Our study fills in this important knowledge gap by **for the first time providing comprehensive and direct physiological evidence** for the MHC-based EET pathways in the *Methanoperedenaceae* ANME. The novel aspects of our study include:

1) First physiological evidence for the direct EET pathway of ANME for metal reduction.

Due to the lack of an ANME pure culture, it is challenging to differentiate the EET behaviours of ANME from the potential electroactivity of other microorganisms in the enrichment consortia. By innovative utilization of fluorescence (based on the autofluorescence and Fe²⁺-selective chemosensory fluorescence) and electron microscopic visualization (Ag (0) deposition from Ag (I) reduction) at the single-cell level, this study determines that ANME can indeed use a direct EET pathway for metal reduction.

2) First identification and *in vivo* characterization of MHCs in ANME. This work applied single-cell TEM visualization (via heme-staining) to identify not only the existence of MHCs in ANME cells, but also their essential role in the EET of ANME. This study also conducted the first *in vivo* electrochemistry study of MHCs in ANME. We demonstrated the first electrochemical identification of MHCs as conduits for EET in living ANME cells by chemically marking hemes with AcMet. This study also used spectroscopic characterization to provide the first spectroscopic information about the active role of extracellular MHCs involved in the EET of ANME to solid electrodes.

3) Comparative metatranscriptomic analyses to reveal different MHC-based EET pathways of ANME. We observed that *Methanoperedens* actively switch between encoded electron transfer pathways to likely allow their flexible use of diverse electron acceptors.

As this study includes a collection of many different methods and results, the novelty of this study may be hidden, and the conclusion of the study may be dispersed. To sharpen and clarify the novelty of this study, we have made the following changes to our introduction and conclusion parts in the revised manuscript.

In Introduction:

Line 69-80. Holmes and colleagues recently used *Methanosarcina acetivorans* as a genetically tractable model microbe to verify that the methanogenic relative of ANME can use membrane-bound cytochrome

as a conduit to transfer electrons to an extracellular electron acceptor³, which has a mechanistic implication for the EET of ANME. MHC genes are more common and abundant within ANME lineages relative to related methanogens^{2,4}. However, investigation of the extracellular electron transfer (EET) mechanisms of ANME has been stymied by the lack of an ANME pure culture. The current knowledge of cytochrome-based EET pathways of ANME is relatively poor and predominantly inferred from metagenomic and metatranscriptomic prediction^{5,6,7}. Up to date, *in vivo* identification, and biochemical and biophysical characterization of MHCs in ANME beyond genomic and transcriptomic analyses are still lacking to verify that cytochromes indeed function as conduits to transfer electrons across the outermost layer to the cell exterior².

In Conclusion:

Line 389-416. Taken together, we present a combination of novel methods to resolve the EET physiology and mechanisms of the ANME ‘*Ca. M. nitroreducens*’. The physiological evidence from fluorescence and single-cell electron microscopic visualization combined revealed the direct EET pathways of ‘*Ca. M. nitroreducens*’. Heme-reactive staining verified the localization of extracellular cytochromes and *in vivo* electrochemical characterization combined with chemically marking hemes with AcMet identified surface cytochromes as the conduits of ‘*Ca. M. nitroreducens*’ for EET. Further spectroscopic analysis verified the active role of MHCs in mediating the electron transfer of ‘*Ca. M. nitroreducens*’. Though this EET pathway has been speculated to be a unifying evolutionary feature of ANME lineages, which generally have a high abundance of genes encoding MHCs^{2,8}, this study for the first time physiologically confirmed the pathway in ANME *Methanoperedenaceae*. Another intriguing finding of this study is that EET pathways of ‘*Ca. M. nitroreducens*’ can be functionally altered by tuning expression levels of various MHCs to accommodate terminal electron transfer to substrates of varying redox potentials. Differential transcriptomic analysis of MHCs reveals that ‘*Ca. M. nitroreducens*’ use different complexes of menaquinone cytochrome type *c* oxidoreductases as well as extracellular MHC conduits, respectively, to likely facilitate the electron transfer out of the menaquinone pool and to different terminal electron acceptors. This electron transfer characteristic resembles what has been described for the EET-capable model microorganism of *Geobacter*, which uses at least two MHC-based complex systems for quinol:cytochrome *c* oxidoreduction step^{9,10}, and multiple putative outer membrane electron MHC conduits for the final electron step to different electron acceptors¹¹. Analogous to the general metabolic versatility of *Geobacter*¹², the tuneable EET pathways may provide ANME with the more metabolic capability than we currently expect, which is likely achieved by the acquisition of MHCs via a hypothetical lateral gene transfer process⁸. Expressing and tuning an array of MHC-based electron transfer pathways could make ANME competitive under diverse fluctuating electron acceptor conditions for niche-specific adaptation. This study takes an important step in our knowledge of the EET pathways of ANME and contributes to our broader understanding of their key roles in linking the global methane cycle with other biogeochemical cycles.

Point-by-point response to Reviewers’ comments

Reviewer #1

1-1. In the manuscript ‘Multi-heme cytochrome mediated extracellular electron transfer by the anaerobic methanotroph ‘Candidatus Methanoperedens nitroreducens’ the authors describe several experiments that provide evidence that Candidatus Methanoperedens nitroreducens uses redox-active cytochromes for iron and electrode reduction. These experiments include differential transcriptomics, electrochemical techniques, Raman spectroscopy and microscopy.

Although I consider the topic of the manuscript interesting, I find that the experiments conducted, and the author’s conclusions lack veracity and fail to reflect their intended findings. Furthermore, this manuscript lacks novelty as the literature already contains evidence, specifically referenced in the manuscript as reference 6, demonstrating that methanotrophic archaea utilize MHC for iron reduction. In my opinion, the conclusions made are questionable

due to their use of a culture stated by the authors as predominantly composed by Ca. M. nitroreducens, rather than a pure culture. This choice undermines the validity of their findings, as there is no evidence to confirm whether the observed results originate from the archaeon in question, or any other organism present in the culture. Indeed, it is described that only 38 and 32% of this species exists in the culture.

Response: Regarding the reviewer's concerns about the novelty of this study: It is true that several key papers explore the use of metal oxides as electron acceptors for *Methanoperedenaceae*^{5,6}. But in these previous papers, the employment of MHCs was simply suggested from metagenomic and metatranscriptomic inferences. We have clarified the novel aspects of our work in the "general response regarding the novelty of this work" described above.

Regarding the reviewer's concerns about using a culture composed predominantly of ANME 'Ca. M. nitroreducens' rather than a pure culture, and it undermines the validity of the findings and veracity of conclusions: we would like to point out that there is currently no pure ANME culture available for any biochemical characterization and potential genetical manipulation to reach the ultimate evidence of MHC-based EET pathways of ANME. That is the reason why the MHC-based EET pathway has relied on genome-based prediction, while biochemical and physiological evidence has rarely been reported.

We agree with the reviewer that it is challenging to perform physiological characterization of ANME within a mixed culture. We believe that we have employed several innovative experiments to distinguish the electroactivity of 'Ca. M. nitroreducens' from other organisms in the community to prove the EET pathways of 'Ca. M. nitroreducens', by making use of the unique identities of 'Ca. M. nitroreducens'. For example:

- 1) By utilizing the characteristic of autofluorescence of 'Ca. M. nitroreducens', we designed a fluorescent method (based on the autofluorescence and Fe²⁺-selective chemosensory fluorescence). With the combination of metatranscriptomic analysis, we revealed the direct EET of ANME for iron reduction.
- 2) Owing to the fortuitous unique morphology of 'Ca. M. nitroreducens' cells, we used single-cell electron microscopic visualization to further confirm the direct EET pathway of 'Ca. M. nitroreducens' (via metal deposition). Combined with TEM visualization of the localization of extracellular cytochromes and the experiment with chemically marking hemes by AcMet, we provided physiological evidence for the MHC-based direct EET pathway of 'Ca. M. nitroreducens'.

We believe we have provided the most convincing evidence so far to resolve the MHC-based direct EET pathway of ANME, given the unavailability of any pure cultures. These are exactly the novel aspects of this study to advance our understanding of the EET pathways of ANME.

1-2. While metatranscriptomic analysis may reveal the predominance of mRNA originating from a particular organism, it does not guarantee that all obtained results, such as those from electrochemistry and Raman spectroscopy, exclusively concern to that organism. This is clearly visible by the results obtained from the biofilm of the BES, where less than 10% of the mRNA was from Ca. M. nitroreducens and more than 23% was from Geobacter.

Response: As explained in our response to comment #1, we have applied specifically designed methods, on the basis of unique properties of 'Ca. M. nitroreducens', to differentiate the

activity of '*Ca. M. nitroreducens*' from other organisms and convince the independent role of ANME in the consortium to do metal reduction with a direct EET pathway.

We agree with the reviewer's specific concerns on bulk results of electrochemistry and Raman spectroscopy to be exclusive of other organisms. It should be pointed out that electroactive bacterium of *Geobacter* was only featured in our long-term operational BESs (Fig. S10), while not exist in any of our other experiments in this study. Our previous study has resolved that the organic (mainly acetate) generated from the intracellular storage and extracellular polymeric substances of '*Ca. M. nitroreducens*', or from cell debris of inoculum biomass, can stimulate the growth and activities of *Geobacter* in the community¹³.

Given the fact that 1) the total expression of MHCs was most predominantly affiliated with '*Ca. M. nitroreducens*' and *Geobacter* on the electrode (Fig. S10); 2) according to multiple pieces of evidence, it was suggested that '*Ca. M. nitroreducens*' highly express MHCs to independently interact with the electrode rather than via obligate syntrophy with *Geobacter* (the detailed justification has shown in Supplementary Text). 3) *Geobacter* was not detected in the inoculum community (Fig. 1E), but only featured in long-term operation of BESs (explained in the last paragraph); 4) the *in vivo* electrochemical characterization of MHCs was conducted in short-term electrochemistry (Fig. 3B), we attributed the MHC redox signals only to '*Ca. M. nitroreducens*' excluding *Geobacter*. This can also be supported by the fact that the midpoint redox potential (E^f) of MHC on the CV profile kept the same over an incubation term of 24 h (Fig. 3B) (otherwise the E^f would be shifted or come with new redox peaks if *Geobacter* grew and expressed MHCs on the electrode).

Similarly for the Raman spectroscopic analysis, we knew that long-term bioelectrochemical incubation will end up with the growth of *Geobacter* in the community to potentially introduce the spectroscopic resolution of MHCs from *Geobacter*. Therefore, we used a sophisticated method with long-term incubation fed with nitrate to form '*Ca. M. nitroreducens*'-dominated electrode biofilm excluding the other electroactive organisms (Fig. S14). This method successfully achieved '*Ca. M. nitroreducens*'-enriched biofilm without other electroactive organisms (Fig. S15). Thus, the MHC Raman spectra were also attributed to '*Ca. M. nitroreducens*' but exclusively to other organisms.

We have highlighted in our manuscript (in both the Discussion (line 306-309) and Method (line 603-608) parts) to address the specific concern on this point for Raman spectroscopic analyses.

We also added the following sentences in the revised manuscript to address this specific concern for electrochemistry results.

Line 265-275. These results combined suggest that *c*-type cytochromes governed the EET of '*Ca. M. nitroreducens*' enrichment culture to directly interact with electrodes. When respiring on electrodes, expression of *c*-type cytochromes was predominantly from '*Ca. M. nitroreducens*' and the electroactive bacterium of *Geobacter* (Fig. S10). *Geobacter* was not detected in the inoculum biomass used for the short-term electrochemical characterizations (Fig. 1E), and appeared only after long-term bioelectrochemical incubations (seen discussion in Supplementary Materials). Thus, the *c*-type cytochromes identified from the short-term *in vivo* electrochemistry analyses most likely originated from '*Ca. M. nitroreducens*'. The *in vivo* electrochemical identification and characterization of *c*-type cytochromes combined with transcriptomic analyses all support MHC-mediated direct electron transfer from '*Ca. M. nitroreducens*' to the electrodes.

In addition to these issues, I also have additional concerns.

1-3. Line 92 – How the authors know if the characterized MHC are important for EET?

Although they identify some of the MHC present in the genome of this organism, it does not mean that the cytochrome will be important for iron reduction.

Response: We agree with the reviewer that the genetic prediction does not definitely mean that MHCs will be used for the EET of ANME for iron reduction. That is indeed one of the main reasons that encouraged us to conduct this study to obtain direct physiological evidence for the MHC-based EET pathways, addressing this important knowledge gap. We made the following modification in the manuscript to clarify the context of testing MHC-based pathways for iron reduction by ANME.

Line 95-103. Previous studies have demonstrated that '*Ca. M. nitroreducens*' enrichment culture, which had been acclimated to nitrate reduction, was able to catalyse iron reduction^{14, 15}. However, it is unknown whether '*Ca. M. nitroreducens*' can reduce iron independently and the EET mechanism it employs for this. High expression of MHCs was previously reported by the related species '*Ca. Methanoperedens ferrireducens*' during iron reduction⁵. As '*Ca. M. nitroreducens*' also encodes a number of MHC genes, it was speculated that it may also express these for direct EET pathways for iron reduction¹⁵. To test this hypothesis, we compared gene expression for a '*Ca. M. nitroreducens*'-dominated culture with either nitrate or iron as the sole terminal electron acceptor, and a series of characterisations on MHCs were conducted.

*1-4. Line 92-93 – How the authors ensure that both reactions (iron and nitrate reduction) are conducted by *Ca. M. nitroreducens*?*

Response: The '*Ca. M. nitroreducens*' was originally enriched for nitrate reduction, which was detailed in our previous paper¹⁶.

Previous papers have demonstrated that '*Ca. M. nitroreducens*' enrichment culture can reduce iron^{14, 15}, but it was unknown whether this species reduced the iron independently or not. This is exactly what we addressed in the current study using combined evidence from fluorescence visualization and metatranscriptomic analysis.

Please see our changes in response to comment 1-3 to clarify this.

1-5. Line 109-110 – Depending on the speciation, iron citrate can enter the cells and be reduced in the periplasm space of Gram-negative bacteria. Since archaeon do not have outer-membrane but a cell envelope, there is no evidence that the reduction of iron citrate occurs at the cell surface. It may also occur at the inner-membrane. Furthermore, the reduction of Fe^{3+} will lead to the formation of Fe^{2+} that is soluble and will diffuse in the media.

Response: We agree with the reviewer that soluble ferric citrate could diffuse through the envelope of '*Ca. M. nitroreducens*', which would be reduced in pseudoperiplasm. We also agree with the reviewer's comment that the formed soluble Fe^{2+} species will diffuse in the media. However, a small amount of ferrous phosphate or ferrous carbonate generated from Fe^{3+} reduction can be bound to enzymes for Fe^{3+} or adsorbed to EPS of the iron reducers¹⁷. This is the mechanism how iron reducers can be tagged with Fe^{2+} -specific fluorescent probes¹⁷. Thus, the Fe^{2+} -specific probe fluorescence only be from '*Ca. M. nitroreducens*' in the consortium, revealing that they use direct EET pathway for iron reaction.

The key message we concluded from the Fe^{2+} -specific fluorescent assay is the direct EET pathway of '*Ca. M. nitroreducens*'. Thus, we've removed the comments on the location of reduction reaction, to make our statements more accurate.

Line 122-124: Collectively, these results indicate that iron reduction was performed by ‘*Ca. M. nitroreducens*’ cells via direct EET.

1-6. Line 125-128 – it is not clear from the text how the MHC were quantified. A reference to the table of genes should be included.

Response: The MHCs were identified by finding the CXXCH amino acid motif in open reading frames (ORFs) of the MAGS in the whole community. This motif represents a heme-binding site. Moreover, for ‘*Ca. M. nitroreducens*’, the MHCs have been previously identified in other papers^{16, 18}, so we checked that our MHCs are the same ORFs as discussed in those papers.

We have edited the paper to now refer to the supplementary tables in which these ORFs can be found. We have also updated the heatmap figure to include gene IDs so that readers can look into the transcription of these differentially expressed MHCs in the supplementary, which may benefit some readers’ own research.

1-7. Line 171 – It is not clear what acetyl methionine will do to cytochromes. The hemes in these proteins are typically hexacoordinated with two histidines, and there is no evidence in the literature that demonstrates that AcMet will have the capacity to replace the strong axial ligand Histidine. Considering that certain enzymes involved in the nitrogen cycle possess hemes with open coordination, it is plausible that AcMet could potentially interact with these cytochromes. However, it is important to note that such interactions would not impact electron transfer to iron or electrodes. This should be discussed in the manuscript.

Response: It was demonstrated that axial ligand coordination reaction of the c-hemes with acetyl methionine (AcMet) can increase the reduction potential ($E^{0'}$) of outer membrane cytochrome *c* (c-Cyts) to cause the formation of a large energy barrier for the electron-exchange process¹⁹. The change of $E^{0'}$ of AcMet-coordinated c-Cyts was also previously demonstrated with purified c-Cyts proteins via biochemical characterizations²⁰.

We agree with reviewer in terms of other possibilities such as the AcMet ligand with heme open coordination. However, in our study, AcMet more likely had ligand coordination reaction with c-Cyts of ‘*Ca. M. nitroreducens*’ culture. Similar to changes that have been observed for AcMet-coordinated c-Cyts, *in vivo* electrochemistry showed that $E^{0'}$ of the surface MHCs of ‘*Ca. M. nitroreducens*’ culture was +390 mV shifted by AcMet coordination (Fig. 3D). This could block the electron flow through electron transport chains of ‘*Ca. M. nitroreducens*’. Indeed, the electron transfer to electrodes for current generation and to silver reduction were both inhibited (Fig. 3C; Fig. S5B).

According to reviewer’s suggestion, we’ve added following discussion in the revised manuscript to clarify what acetyl methionine will do to cytochromes.

Line 194-201. The axial ligand coordination reaction of the *c*-hemes with AcMet has been demonstrated to increase the reduction potential ($E^{0'}$) of *c*-type cytochromes to cause the formation of a large energy barrier for the electron transfer process^{19, 20}. This AcMet coordination could thus block the electron flow through electron transport chains of ‘*Ca. M. nitroreducens*’ for the observed inhibition of silver reduction. These results collectively suggest that ‘*Ca. M. nitroreducens*’ uses direct EET pathways for Ag (I) reduction to Ag (0) by using *c*-type cytochromes localized on the cell surface or anchored to EPS surface as electrical conduits.

Line 254-265: To further examine whether the redox-active species governing the direct EET pathway of '*Ca. M. nitroreducens*' enrichment culture were putative *c*-type cytochromes, we performed axial-coordination reaction on the heme groups of *c*-type cytochromes under living conditions by using the specific binding affinity of AcMet. The E^f shifted to around +390 mV (Fig. 3C), in contrast to -20 mV observed prior to the AcMet addition. The increase of redox potential as a response to AcMet coordination was also previously observed for the purified *c*-type cytochromes proteins²⁰, and OMCs in living electroactive *Shewanella* cells.¹⁹ It caused the formation of a large energy barrier for the electron-exchange process of electroactive bacteria (*Shewanella* and *Geobacter*)^{19,21}. Congruously, the addition of 100 mM AcMet also resulted in complete suppression of the catalytic current density of the '*Ca. M. nitroreducens*' enrichment culture incubated at 0.4 V vs SHE upon (Fig. 3D).

1-8. Line 177 - TEM images – the authors describe that the nanoparticles are localized at the outer surface of *Ca. M. nitroreducens*, but from the figure it seems that they are at the EPS matrix and not at the cell surface. Given the figures observed with DAB, it seems that the cell surface is much more near the contours of the cell. Furthermore, it is not clear how the authors know that these are *Ca. M. nitroreducens* cells. This should be clear in the text.

Response: We agree with the reviewer that we cannot exclude the possibility that '*Ca. M. nitroreducens*' has surface-associated EPS and AgNPs may be deposited there. However, we also see clear evidence that some AgNPs are deposited on the envelope surface of '*Ca. M. nitroreducens*' (this is more clearly viewed from Fig. S6). Moreover, the observation of precipitation of AgNPs in EPS does not collide with the conclusion of the study, as EPS of EET-capable microorganisms is an electroactive transient media to transfer electrons outward from cell surface across the EPS layer²². Thus, it is not surprising see the localization of AgNPs on the surface of EPS. We have expanded the relevant text to address the reviewers concerns and hope that it is now clearer.

Line 185-189: Thin sectioning TEM showed AgNPs were deposited either on outer envelope surface or associated-extracellular polymeric substances (EPS) of '*Ca. M. nitroreducens*' cells (Fig. 2E, F; Fig. S6 A, B, E). The observation of AgNPs on the outer surface of cells are similar to the precipitation of Ag (0) on the outer membrane of *G. sulfurreducens* and *Thermincola potens* from their respiratory Ag (I) reduction^{23, 24}. Meanwhile, as the EPS of EET-capable microorganisms is an electroactive transient media to transfer electrons outward across the EPS layer²², it is rational to see the precipitation of AgNPs also on the surface of EPS.

Line 199-201: These results collectively suggest that '*Ca. M. nitroreducens*' uses direct EET pathways for Ag (I) reduction to Ag (0) with *c*-type cytochromes localized on the cell surface or anchored to EPS surface as electrical conduits.

Regarding the reviewer's questions on how we know these cells in TEM images are '*Ca. M. nitroreducens*', we identify them based on their unique morphologies in comparison to cells of other populations in the consortium, which has been clarified via extensive FISH analyses of the same community in our previous work^{16, 18}: 1) they normally appear as irregular cocci that are larger than the cells of other populations, and typically form sarcina-like clusters¹⁶ (Fig. 2A, B, C, D, E, G; Fig. S6A, C); 2). They are also the only population in the consortium that store polyhydroxyalkanoate granules¹⁸ (Fig. 2F, H; Fig. S6 B, D). We've added the following sentences in the revised manuscript to clarify this.

Line 521-526: Single '*Ca. M. nitroreducens*' cells were distinguished in TEM images according to their unique morphology, based on extensive fluorescence *in situ* hybridization (FISH) characterization of the consortium: 1) they normally grow with irregular cocci shape at larger size to other populations, and they are typically found as sarcina-like clusters¹⁶; 2) they are the only population in the consortium with detectable polyhydroxyalkanoate (PHA) granule storage¹⁸, which can also be observed with TEM.

We also noted this in the figure caption for easier reading:

Line 204-205: TEM images showing heme reactivity of '*Ca. M. nitroreducens*' cells (identified by their distinctive morphology and size) in the iron-reducing incubation.

1-9. Line 309-310 – *If the authors are trying to identify MHC, they should know that all MHC have a signal peptide. While not all signal peptides undergo cleavage in these proteins, they are required to translocate the proteins outside the cytoplasm where the heme maturation process occurs. While in bacteria, the maturation occurs at the inner-membrane level towards the periplasmic space, in archaea this occurs towards the cell envelope.*

Response: We agree with the reviewer MHCs require a signal peptide for translocation and have deleted the sentence and “Putative Signal Peptide” in Fig. 4 to avoid confusion.

1-10. Line 316-317 – *The results obtained by the authors relative to the expression of the different cytochromes during different metabolic processes cannot be compared given that the number and type of other species in the cultures are different. The difference of the expression of the different proteins are not only due to the different metabolic processes but may also be influenced by the different species and compounds existent in the culture.*

Response: We agree that there are caveats to the comparison of metaT data across samples, but not that such comparisons are not informative. To clarify our methodology, the analysis chosen in our manuscript is suitable for comparison of the MHC expression by '*Ca. M. nitroreducens*' across samples. For DE analysis, gene expression for '*Ca. M. nitroreducens*' is scaled by DESeq2's method 'median of ratios': gene expression counts are divided by sample-specific size factors determined by median ratio of gene counts relative to geometric mean per gene. This normalises the data to account for the higher or lower total expression of '*Ca. M. nitroreducens*' in the MetaT of some samples. To visualise MHC expression in the heatmap figure, a regularised log transformation on the normalised counts was performed and then this was converted into a Z-score across the rows (genes). We can observe the log scale changes in expression for each gene across the conditions, but the analysis is not intended for comparison of gene expression within each sample, as the log fold change is calculated for each row individually. We agree that an observed increase in expression of an MHC associated with the introduction of an electron acceptor does not guarantee its direct involvement in the reduction of that electron acceptor. But we believe it is enough to hypothesize its involvement - which can importantly inform further empirical study.

Reviewer #2

2-1. *The study of Zhang et al. characterizes the physiology of Candidatus Methanoperedens nitroreducens with state-of-the-art spectroscopic methods, electrochemical procedures and metatranscriptomics.*

The experiments are difficult since only enrichment cultures of these archaea are available that predominantly grow in syntrophy with anammox bacteria.

Response: We appreciate that the reviewer understands the difficult nature of all the experiments conducted with this uncultured archaeon.

General comments

2-2. *The study is a collection of many different methods, whereby the focus lays on the role of the multi-heme c type cytochromes. The experiments are well-conducted and involve custom-made bioelectric set ups to study methane oxidation coupled to the transfer of the electrons to electrodes.*

The collection of data reported here should be published since they can be valuable to the specialist in the field of electron-conducting proteins and for the field of anaerobic methane oxidation in general.

While a good set of data has been acquired, no big fundamentally new discovery has been made, and it is difficult to draw a short, relevant conclusion from the study. My judgement of “collection of data” is substantiated by the fact that the authors write often “further support”, “confirming”, “further supporting”, “further insight” etc. throughout the manuscript. It could be considered to publish this work in a more specialized journal.

Response: We also appreciate the reviewer’s credits on the experiments themselves as well the significance of this work to different scientific communities.

Regarding the reviewer’s comment on no fundamentally new discoveries and relevant conclusions, we have justified the novelties and significance of this study in the “general response regarding the novelty of this work:” as described above.

Due to the lack of pure culture of ANME, we backed up each conclusion by multiple pieces of evidence from different experiments. That is the main reason why these terms like ‘further supporting/confirming’ noted by the reviewer were frequently used, since we described the outcomes of multiple experiments in sequence.

Selected comments by line number:

2-3. - line 34-37: *This part should be elaborated more in the main manuscript, if this is the conclusion of the abstract*

Response: As suggested by the reviewer, we have added more discussion in the main manuscript regarding this point.

Line 371-378: The distinct expression of extracellular MHCs indicates that ‘*Ca. M. nitroreducens*’ may use different putative outer surface electron conduits to transfer electrons to different extracellular electron acceptors or to interdependent species. This is similar to the metabolic characteristic of a model EET-capable microorganism, i.e., *Geobacter*, which was also identified to use different out membrane conduits to respire a wide array of extracellular substrates¹¹. The distinctive MHC expression patterns collectively suggested that ‘*Ca. M. nitroreducens*’ can use different MHC-based pathways to allow the use of different electron acceptors.

Line 399-416: Another intriguing finding of this study is that EET pathways of ‘*Ca. M. nitroreducens*’ can be functionally altered by tuning expression levels of various MHCs to accommodate terminal electron transfer to substrates of varying redox potentials. Differential transcriptomic analysis of MHCs reveals that ‘*Ca. M. nitroreducens*’ use different complexes of menaquinone cytochrome type *c* oxidoreductases as well as extracellular MHC conduits, respectively, to likely facilitate the electron transfer out of the menaquinone pool and to different terminal electron acceptors. This electron transfer characteristic resembles what has been described for the EET-capable model microorganism of *Geobacter*, which uses at least two MHC-based complex systems for quinol:cytochrome *c*

oxidoreduction step^{9, 10}, and multiple putative outer membrane electron MHC conduits for the final electron step to different electron acceptors¹¹. Analogous to the general metabolic versatility of *Geobacter*,¹² the tuneable EET pathways may provide ANME with the more metabolic capability than we currently expect, which is likely achieved by the acquisition of MHCs via a hypothetical lateral gene transfer process⁸. Expressing and tuning an array of MHC-based electron transfer pathways could make ANME competitive under diverse fluctuating electron acceptor conditions for niche-specific adaptation. This study takes an important step in our knowledge of the EET pathways of ANME and contributes to our broader understanding of their key roles in linking the global methane cycle with other biogeochemical cycles.

2-4. - line 73 “seldomly conducted”: please cite where it has been conducted

Response: The paper by Chadwick et al., (2021) has now been cited in the revised manuscript.

2.5 - line 86 ff: even though the methods come later, it would be nice to know here already what we are talking about. That it is an enrichment culture coupled to ammonium oxidation (right?) that is switched to the (perhaps artificial) metabolism of metal reduction.

Response: We agree and have modified the sentence accordingly.

Line 95-103. Previous studies have demonstrated that ‘*Ca. M. nitroreducens*’ enrichment culture, which had been acclimated to nitrate reduction, was able to catalyse iron reduction^{14, 15}. However, it is unknown whether ‘*Ca. M. nitroreducens*’ can reduce iron independently and the EET mechanism it employs for this. High expression of MHCs was previously reported by the related species ‘*Ca. Methanoperedens ferrireducens*’ during iron reduction⁵. As ‘*Ca. M. nitroreducens*’ also encodes a number of MHC genes, it was speculated that it may also express these for direct EET pathways for iron reduction¹⁵. To test this hypothesis, we compared gene expression for a ‘*Ca. M. nitroreducens*’-dominated culture with either nitrate or iron as the sole terminal electron acceptor, and a series of characterisations on MHCs were conducted.

2.6 - line 89 “suggested”: didn’t they obtain a culture where this process really happens?

Response: They did use the same culture to observe the phenomenon of iron reduction. However, they did not investigate the mechanism of how iron was reduced.

2.7 - line 111 “to better understand”: in the conclusion it should be more clearly stated what is now clearer thanks to this study.

Response: We have more clearly expressed the conclusion of this study.

Line 389-396: Taken together, we present a combination of novel methods to resolve the EET physiology and mechanisms of the ANME ‘*Ca. M. nitroreducens*’. The physiological evidence from fluorescence and single-cell electron microscopic visualization combined revealed the direct EET pathways of ‘*Ca. M. nitroreducens*’. Heme-reactive staining verified the localization of extracellular cytochromes and *in vivo* electrochemical characterization combined with chemically marking hemes with AcMet identified surface cytochromes as the conduits of ‘*Ca. M. nitroreducens*’ for EET. Further spectroscopic analysis verified the active role of MHCs in mediating the electron transfer of ‘*Ca. M. nitroreducens*’.

2.8 - line 131 (fig. 1): *The quality of the figure is not sufficient. I cannot read panel F, for example since too small font anyway and it appears blurry. Panel B-D: where are the partner bacteria? Are those mostly archaea? I have missed a comment on that, or it is not explained.*

Response: We have increased the quality of the Fig. 1. Seen in the revised manuscript.

As the partner bacteria do not have either F₄₂₀ autofluorescence, or positive fluorescent Fe²⁺-specific FeRhoNox staining, they can hardly be visualized with any fluorescence in Fig. 1B-D. But they do accompany archaea cells in the consortium, which can be confirmed with phase-contrast micrographs (Fig. S3G, J). We clarified this in the revised manuscript.

Line 114-118: FeRhoNox positive cells almost entirely overlapped with the '*Ca. M. nitroreducens*' (Fig. 1D; congruency = 94.1 ± 2.5 %, n = 20), while cells of other populations in the consortium showed fluorescence similar to the negative control (Fig. S3G, J), suggesting the former population was responsible for the bulk of the observed iron reduction in the system (Fig. S3E-J).

2.9 - line 162: *How specific is this Ag⁺ method? It seems that Ag⁺ is quite reactive anyway. Should be discussed how indicative this is.*

Response: It has been demonstrated that EET-capable microorganisms can enzymatically reduce Ag(I), via a mechanism involving *c*-type cytochromes, precipitating extracellular nanoscale Ag(0)^{23, 24}. Thus, it can be used as a physiological method to investigate the *c*-type cytochromes-based EET pathways of electroactive microorganisms, with the combination microscopic images.

According to the reviewer's suggestion, we've added some context in the revised manuscript on the Ag-reducing method.

Line 177-183. The nitrate-fed '*Ca. M. nitroreducens*' enrichment culture was also incubated with Ag (I) to examine whether they can also mediate extracellular Ag (I) reduction. It has been demonstrated that EET-capable microorganisms can enzymatically reduce Ag(I), via the mechanism involving *c*-type cytochromes, resulting in precipitation of nanoscale Ag(0) on the cell surface^{23, 24}. As '*Ca. M. nitroreducens*' cells can be distinguished from other populations based on their size and spatial arrangement, examining the potential deposition of the AgNPs on their cell surface would further support their ability for direct EET for metal reduction.

2.10 - line 164 "*further support*": *this is throughout the paper. It is ok like this, but it highlights that the whole study is a collection of data that all point into this direction, but not a solid finding.*

Response: We have clarified this point in our response to comment 2-2. We also use this phrasing to be conservative in our conclusions.

2.11 - line 236: *This "AcMet" seems to appear without warning. I'm not aware of this method. Would be good to briefly discuss what this is and what it tells.*

Response: We've added corresponding sentences in the revised manuscript to clarify the principle of this method.

Line 194-201. The axial ligand coordination reaction of the *c*-hemes with AcMet has been demonstrated to increase the reduction potential (E^0) of *c*-type cytochromes to cause the formation of a large energy barrier for the electron transfer process^{19, 20}. This AcMet coordination could thus block the electron flow

through electron transport chains of '*Ca. M. nitroreducens*' for the observed inhibition of silver reduction.

Line 254-266: To further examine whether the redox-active species governing the direct EET pathway of '*Ca. M. nitroreducens*' enrichment culture were putative *c*-type cytochromes, we performed axial-coordination reaction on the heme groups of *c*-type cytochromes under living conditions by using the specific binding affinity of AcMet. The E^f shifted to around +390 mV (Fig. 3C), in contrast to -20 mV observed prior to the AcMet addition. The increase of redox potential as a response to AcMet coordination was also previously observed for the purified *c*-type cytochromes proteins²⁰, and OMCs in living electroactive *Shewanella* cells.¹⁹ It caused the formation of a large energy barrier for the electron-exchange process of electroactive bacteria (*Shewanella* and *Geobacter*)^{19,21}. Congruously, the addition of 100 mM AcMet also resulted in complete suppression of the catalytic current density of the '*Ca. M. nitroreducens*' enrichment culture incubated at 0.4 V vs SHE upon (Fig. 3D). These results combined suggest that *c*-type cytochromes governed the EET of '*Ca. M. nitroreducens*' enrichment culture to directly interact with electrodes.

2.12 - line 349: *I guess this is the "conclusion", which highlights that there is not so much specific conclusion from this study (even though the study is valuable for the scientific community).*

Response: We acknowledge that the specific conclusions may not have been clear in the original version. We have clarified the new discoveries and main conclusions of this study in our "general response regarding the novelty of this work" and revised the conclusion section to clarify and highlight the specific conclusions.

Reviewer #3

Comments to Zhang et al.

3-1. *The manuscript entitled "Multi-heme cytochrome mediated extracellular electron transfer by the anaerobic methanotroph 'Candidatus Methanoperedens nitroreducens'" by Zhang et al. presents evidence from various visualization, electrochemical and spectroscopic characterization, and genomic analysis to support the hypothesis that extracellular multi-heme cytochromes (MHCs) play a critical role in the EET pathways of "Ca. M. nitroreducens". The hypothesis was frequently invoked in both microbiology- and biogeochemistry-centered studies of metal-dependent ANME. However, evidence beyond genomic data were unfortunately lacking. This work therefore should fill an important knowledge gap in this regard. The experiments were well-designed, the collected data extensive and convincing, and the manuscript was well-written and precise. I have only a few minor comments to the authors at some places that need to pay attention to.*

Response: We appreciate the reviewer's recognition of the significance and value of our study.

Minor comments

3-2. *Lines 105-108: this sentence is confusing. What does the phrase "in the present of Fe3+ but without incubation" mean? Does it mean for the "Ca. M. nitroreducens" culture it was too soon after the Fe3+ addition to have activity? Also, it is unclear what "in the absence of staining" mean.*

Response: As clarified in the M&M, the phrase “in the present of Fe³⁺ but without incubation” means that the ‘*Ca. M. nitroreducens*’ enrichment culture was mixed with Fe³⁺ and immediately stained (i.e. without the iron-reducing incubation process) with FeRhoNoxTM; the phrase “in the absence of staining” means The ‘*Ca. M. nitroreducens*’ enrichment culture was collected at the end of iron-reducing incubation but visualized without FeRhoNoxTM staining.

We’ve rephrased the whole sentence to make it clear.

Line 118-124: Giving confidence in the specificity of the method, FeRhoNox positive signal was not observed for 1) the Fe³⁺-amended ‘*Ca. M. nitroreducens*’ enrichment culture that was immediately stained with FeRhoNox for visualization (i.e. without incubation process for Fe³⁺ reduction) (Fig. S3A, B), or for 2) ‘*Ca. M. nitroreducens*’ enrichment culture that was incubated for Fe³⁺ reduction but in the absence of staining (Fig. S3C, D).

3-3. *This manuscript reports some sequencing data of both metagenome and metatranscriptome. There are also some MAGs were generated in the study. These data should be publicly accessible in some way. Probably adding a separate data availability section can address this issue.*

Response: We agree. The data has been deposited into relevant public databases and accessions numbers have now been included.

Line 727-728: All raw metagenomic and metatranscriptomic reads are archived in NCBI database with BioProject ID PRJNA1003560.

3-4. *For Fig. 1F: I first can only see the blue bars, but not the others. I think it is important to clarify that this panel share the same figure key with Fig. 1E, if my understanding is correct.*

Response: We’ve modified the figure to increase its quality. We’ve also clarified in the figure caption that Fig. 1E and Fig. 1F share the same figure legend.

References

1. McGlynn SE, Chadwick GL, Kempes CP, Orphan VJ. Single cell activity reveals direct electron transfer in methanotrophic consortia. *Nature* **526**, 531-535 (2015).
2. Chadwick GL, *et al.* Comparative genomics reveals electron transfer and syntrophic mechanisms differentiating methanotrophic and methanogenic archaea. (2021).
3. Holmes DE, *et al.* A membrane-bound cytochrome enables *Methanosarcina acetivorans* to conserve energy from extracellular electron transfer. *MBio* **10**, e00789-00719 (2019).
4. Kletzin A, Heimerl T, Flechsler J, van Niftrik L, Rachel R, Klingl A. Cytochromes c in Archaea: distribution, maturation, cell architecture, and the special case of *Ignicoccus hospitalis*. *Frontiers in microbiology* **6**, 439 (2015).
5. Cai C, *et al.* A methanotrophic archaeon couples anaerobic oxidation of methane to Fe (III) reduction. *The ISME journal* **12**, 1929-1939 (2018).
6. Leu AO, *et al.* Anaerobic methane oxidation coupled to manganese reduction by members of the Methanoperedenaceae. *The ISME Journal*, 1-12 (2020).

7. Krukenberg V, *et al.* Gene expression and ultrastructure of meso - and thermophilic methanotrophic consortia. *Environmental microbiology* **20**, 1651-1666 (2018).
8. Leu AO, McIlroy SJ, Ye J, Parks DH, Orphan VJ, Tyson GW. Lateral gene transfer drives metabolic flexibility in the anaerobic methane-oxidizing archaeal family methanoperedenaceae. *MBio* **11**, e01325-01320 (2020).
9. Levar CE, Chan CH, Mehta-Kolte MG, Bond DR. An inner membrane cytochrome required only for reduction of high redox potential extracellular electron acceptors. *MBio* **5**, e02034-02014 (2014).
10. Levar CE, Hoffman CL, Dunshee AJ, Toner BM, Bond DR. Redox potential as a master variable controlling pathways of metal reduction by *Geobacter sulfurreducens*. *The ISME journal* **11**, 741-752 (2017).
11. Jiménez Otero F, Chan CH, Bond DR. Identification of different putative outer membrane electron conduits necessary for Fe (III) citrate, Fe (III) oxide, Mn (IV) oxide, or electrode reduction by *Geobacter sulfurreducens*. *Journal of bacteriology* **200**, 10.1128/jb. 00347-00318 (2018).
12. Methe B, *et al.* Genome of *Geobacter sulfurreducens*: metal reduction in subsurface environments. *Science* **302**, 1967-1969 (2003).
13. Zhang X, *et al.* Polyhydroxyalkanoate-driven current generation via acetate by an anaerobic methanotrophic consortium. *Water Research*, 118743 (2022).
14. Ettwig KF, Zhu B, Speth D, Keltjens JT, Jetten MS, Kartal B. Archaea catalyze iron-dependent anaerobic oxidation of methane. *Proceedings of the National Academy of Sciences* **113**, 12792-12796 (2016).
15. Cai C, *et al.* Response of the Anaerobic Methanotrophic Archaeon Candidatus “*Methanoperedens nitroreducens*” to the Long-Term Ferrihydrite Amendment. *Frontiers in Microbiology* **13**, 799859 (2022).
16. Haroon MF, *et al.* Anaerobic oxidation of methane coupled to nitrate reduction in a novel archaeal lineage. *Nature* **500**, 567-570 (2013).
17. Gan C, *et al.* Visualizing and Isolating Iron-Reducing Microorganisms at the Single-Cell Level. *Applied and Environmental Microbiology* **87**, e02192-02120 (2020).
18. McIlroy SJ, *et al.* Anaerobic methanotroph ‘Candidatus *Methanoperedens nitroreducens*’ has a pleomorphic life cycle. *Nature Microbiology* **8**, 321-331 (2023).
19. Nakamura R, Kai F, Okamoto A, Newton GJ, Hashimoto K. Self - constructed electrically conductive bacterial networks. *Angewandte Chemie International Edition* **48**, 508-511 (2009).
20. Battistuzzi G, Borsari M, Cowan JA, Ranieri A, Sola M. Control of cytochrome c redox potential: axial ligation and protein environment effects. *Journal of the American Chemical Society* **124**, 5315-5324 (2002).
21. Malvankar NS, Tuominen MT, Lovley DR. Lack of cytochrome involvement in long-range electron transport through conductive biofilms and nanowires of *Geobacter sulfurreducens*. *Energy & Environmental Science* **5**, 8651-8659 (2012).

22. Xiao Y, *et al.* Extracellular polymeric substances are transient media for microbial extracellular electron transfer. *Science advances* **3**, e1700623 (2017).
23. Law N, Ansari S, Livens FR, Renshaw JC, Lloyd JR. Formation of nanoscale elemental silver particles via enzymatic reduction by *Geobacter sulfurreducens*. *Applied and environmental microbiology* **74**, 7090-7093 (2008).
24. Carlson HK, *et al.* Surface multiheme c-type cytochromes from *Thermincola potens* and implications for respiratory metal reduction by Gram-positive bacteria. *Proceedings of the National Academy of Sciences* **109**, 1702-1707 (2012).